# Best-of-∞ – Asymptotic Performance of Test-Time LLM Ensembling

**Junpei Komiyama**
Mohamed bin Zayed University of Artificial Intelligence
New York University / RIKEN AIP
junpei@komiyama.info

**Daisuke Oba**
Institute of Science Tokyo
daisuke.oba@nlp.comp.isct.ac.jp

**Masafumi Oyamada**
NEC Corporation
stillpedant@gmail.com

## Abstract

We study best-of-$N$ for large language models (LLMs) where the selection is based on majority voting. In particular, we analyze the limit $N \to \infty$, which we denote as best-of-$\infty$. While this approach achieves impressive performance in the limit, it requires an infinite test-time budget. To address this, we propose an adaptive generation scheme that selects $N$ based on answer agreement, thereby efficiently allocating inference-time computation. Beyond adaptivity, we extend the framework to weighted ensembles of multiple LLMs, showing that such mixtures can outperform any individual model. The optimal ensemble weighting is formulated and efficiently computed as a mixed-integer linear program. Extensive experiments demonstrate the effectiveness of our approach. Our code is available at https://github.com/jkomiyama/BoInf-code-publish/.

## 1 Introduction

The last few years have witnessed remarkable advancements in large language models (LLMs), in their industrial successes including closed models such as Gemini (Gemini Team, 2025), GPT (OpenAI, 2023), and Claude (Anthropic, 2025) as well as open-weight models such as Llama (Llama Team, 2024), Deepseek (DeepSeek-AI, 2025), Qwen (Qwen Team, 2025; Ye et al., 2025; Cheng et al., 2025) and many others including Liu et al. (2023); Almazrouei et al. (2023); Gao et al. (2023); Jiang et al. (2023a); Biderman et al. (2023); BigScience Workshop (2023); OpenAI (2025); Wang et al. (2025b); NVIDIA (2025); Abdin et al. (2025); Ji et al. (2025); LG AI Research (2025). One of the largest interests in the realm of LLMs is on their ability to perform complex reasoning tasks. A breakthrough in the reasoning of LLMs was the introduction of chain-of-thought prompting (Wei et al., 2022; Kojima et al., 2023), which allows models to generate intermediate reasoning steps before arriving at an answer. Instruction-tuned LLMs optimized to generate longer chains of thought have drastically increased performance in these tasks (Muennighoff et al., 2025).

Spending more computational resources at test time, in particular by generating multiple answers, leads to more reliable inference (Snell et al., 2025; Brown et al., 2024). A simple yet effective strategy is the best-of-$N$ (BoN) approach, where we generate $N$ answers and select the best one based on some criteria. There are several ways to implement the BoN strategy. One common approach is to use a reward model to select the best answer (Uesato et al., 2022; Rafailov et al., 2023; Wan et al., 2024; Dong et al., 2024; Liu et al., 2024; Wang et al., 2024; Wu et al., 2025) or asking LLM to choose a preferable answer (Mahan et al., 2024; Son et al., 2024; Guo et al., 2025; Chen et al., 2025a). Another approach is majority voting (Wang et al., 2023) in which the most frequent answer is selected.

Despite its simplicity, majority voting has several advantages. First, it does not require any additional modeling or further text generation. Second, compared with other methods, majority voting is robust to reward hacking and benefits from additional generations with minimal risk, unlike reward-based models where increasing $N$ can lead to overfitting (Huang et al., 2025). Third, for reasoning tasks,

majority vote is reported to be very effective (Chen et al., 2024a). Across datasets, with a few exceptions (Chen et al., 2024b), majority voting performance generally increases with $N$ (Figure 1).

While we desire to achieve such Best-of-$N$ performance of $N \to \infty$, which we call best-of-$\infty$ performance, it requires an infinite number of generations (samples), which is infeasible in real-world scenarios. Yet, for the same test-time budget, we can utilize the available budget more effectively. As shown in Figure 2, we can generate samples adaptively until we determine the majority with some confidence level. We introduce a principled method to determine when to stop generating answers and when to continue using Bayesian modeling (Section 2).

Our scheme can be naturally extended to ensembles of multiple LLMs. Importantly, ensemble majority voting can naturally benefit from complementarity. For example, in the AIME2025 dataset, the best-of-$\infty$ performance of GPT-OSS-20B (OpenAI, 2025) and Nemotron-Nano-9B-v2 (NVIDIA, 2025) are 90.0% and 73.0%, respectively, but their ensemble achieves 93.3%. A weak LLM can contribute to the ensemble if it has complementary strengths.

A key theoretical contribution of this work is the formulation of optimal ensemble weighting as a tractable optimization problem. We show that finding the optimal weight vector that maximizes best-of-$\infty$ accuracy can be reduced to a mixed-integer linear program (MILP) (Section 3). This formulation is enabled by considering the asymptotic limit: while optimizing weights for finite $N$ requires enumerating an exponentially large number of answer combinations, the best-of-$\infty$ framework yields a polytope structure that allows efficient optimization via standard MILP solvers. To our knowledge, this is the first work to provide a computationally tractable method for finding provably optimal ensemble weights in the context of LLM majority voting.

Finally, we evaluate the performance of the proposed method (Section 4). Our experimental results include 11 instruction-tuned LLMs and four heavy-reasoning problem sets (AIME2024, AIME2025, GPQA-DIAMOND, MATH500), with at least 80 generations for each LLM–problem set combination. This represents a significantly larger scale of test-time computation than prior work. We demonstrate that the MILP-optimized ensemble weights consistently outperform both uniform weighting and single-model selection across all benchmarks. We release our generation results for subsequent research. Related work is discussed in Appendix B.

## 2 BEST-OF-$\infty$ IN FINITE SAMPLES

While Best-of-$\infty$ defines an idealized best-of-$N$ ensemble in the limit $N \to \infty$, its literal realization would require unbounded test-time compute. We now develop a finite-sample procedure that closely tracks this limit. Our core idea is to adaptively samples (i.e., ask LLM to generate the answers) until we are sure the population majority vote with a desired confidence level. In other words, we aim to terminate the answer generation process as soon as sufficient statistical evidence has been obtained to support the conclusion that the currently most frequent response corresponds to the true majority, which allows different number of $N$ across problems. A distinctive challenge of this problem lies in the fact that the support of the answer distribution generated by large language models (LLMs) is

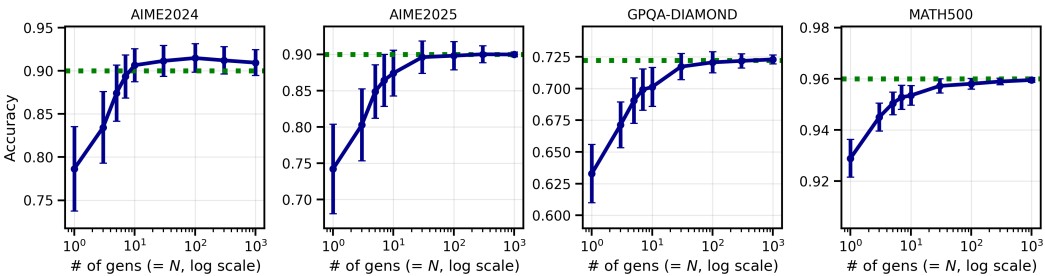

Figure 1: Accuracy of Best-of-$N$ with majority voting as a function of $N$ (GPT-OSS-20B (Medium)) with four datasets (Maxwell-Jia, 2024; OpenCompass, 2025; Rein et al., 2023; Hendrycks et al., 2021). Green line indicates the asymptotic accuracy of $N \to \infty$. For each problem, BoN benefits from increasing $N$, at least from $N = 10^1$ to $10^2$.

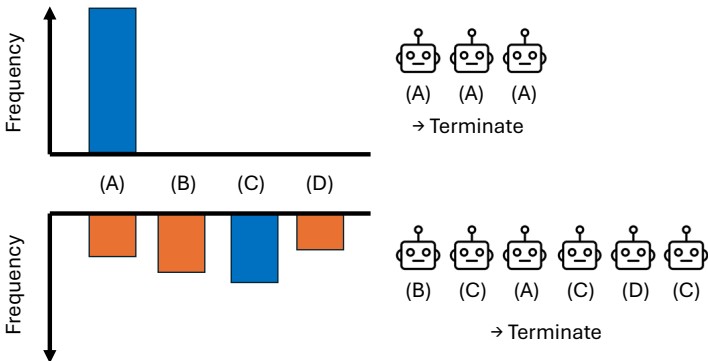

Figure 2: An illustration of adaptive sampling (Algorithm 1). The histogram shows the distribution of answers generated by an LLM for a single problem. Each answer generation can be viewed as a sample from the underlying distribution. Blue indicates the most frequent answer, and orange indicates the others. In the top example, three generations agree, so sampling stops. In the bottom example, more samples are needed to determine the majority. This maximizes the accuracy under a given compute budget. Confidence in the majority is based on the Bayes factor.

---

**Algorithm 1** Approximated Best-of-$\infty$: Determining answer for single problem

---

**Require:** Maximum samples $N_{\max}$, concentration parameter $\alpha$, Bayes factor threshold $B$.
 1: **for** $n = 1, 2, \ldots$ **do**
 2:     **if** we use LLM Ensemble (Section 3) **then**
 3:         Choose LLM with probability $\{w_i\}_{i \in \mathcal{K}}$.
 4:     **end if**
 5:     Ask the LLM for the answer of the problem to obtain answer.
 6:     **if** $n = N_{\max}$ or $\mathrm{BF}(n) \geq B$ **then**
 7:         **break**
 8:     **end if**
 9: **end for**
10: **return** The most frequent answer.

---

unknown. For instance, in one case an LLM may produce two candidate answers, such as 42 with probability 70% and 105 with probability 30%, whereas in another case it may yield four distinct outputs, such as 111 with probability 40%, 1 with probability 25%, 2 with probability 20%, and 702 with probability 15%. Given such uncertainty in the variation of generated responses, a particularly well-suited approach is to employ nonparametric Bayesian modeling. In particular, we adopt a Dirichlet process $\mathrm{DP}(H, \alpha)$ prior over the answer space that captures the unknown distribution of answers. Here, $H$ is a base distribution[1] over the answer space, and $\alpha > 0$ is a concentration parameter that controls the likelihood of generating new answers. Intuitively speaking, $\alpha$ is the strength of the prior belief in the existence of new answers. Assume that, at round $n$, we observe $s(n)$ different answer $A_1, A_2, \ldots, A_{s(n)}$ with corresponding counts $N_1 \geq N_2 \geq N_3 \cdots \geq N_{s(n)}$. Then, the posterior distribution is

$$\mathrm{DP}\left( \underbrace{\frac{\alpha}{\alpha + n} H}_{\text{base distribution}} + \underbrace{\frac{1}{\alpha + n} \sum_{j=1}^{s(n)} N_j \delta_{A_j}}_{\text{empirical distribution}}, \alpha + n \right). \tag{1}$$

---

[1]The base distribution can have a possibly infinite support, such as all possible integers. For some tasks, such as GPQA, the answer is given in a finite domain (e.g., A, B, C, D), and thus the base distribution is of a finite support. In such cases, Dirichlet process is exactly the same as the Dirichlet distribution. The advantage of the Dirichlet process is to unify the treatment for both finite and infinite answer spaces, as well as having some regularization with a hyperparameter $\alpha$.

The first argument of the posterior above states that the posterior is increasingly concentrated around the observed answers as more data is collected.

We use the Bayes factor (Jeffreys, 1935; Good, 1967; Kass & Raftery, 1995; Lindon & Malek, 2022) to measure the evidence of true majority.[2]

Formally, we define the hypotheses as follows:

$$H_0 : \text{The most frequent answer } A_1 \text{ is not the true majority.} \tag{2}$$

$$H_1 : \text{The most frequent answer } A_1 \text{ is the true majority.} \tag{3}$$

and define the Bayes factor (BF), which quantifies the strength of evidence in the data for $H_1$, as

$$\text{BF} := \frac{\mathbb{P}(\mathcal{D}(n)|H_1)}{\mathbb{P}(\mathcal{D}(n)|H_0)}, \tag{4}$$

where $\mathcal{D}(n)$ is the observed data so far. Here, $\mathbb{P}(\mathcal{D}(n)|H_1), \mathbb{P}(\mathcal{D}(n)|H_0)$ are the evidence (marginal likelihood) based on the observed data. Then, the Bayes factor of equation 4 can be computed as follows:

$$\text{BF}(n) := \frac{\mathbb{P}(\mathcal{D}(n)|H_1)}{\mathbb{P}(\mathcal{D}(n)|H_0)} = \frac{\mathbb{P}(H_1|\mathcal{D}(n))}{\mathbb{P}(H_0|\mathcal{D}(n))} \cdot \frac{\mathbb{P}(H_0)}{\mathbb{P}(H_1)} \quad \text{(Bayes' theorem)} \tag{5}$$

$$\approx s(n) \frac{\mathbb{P}(H_1|\mathcal{D}(n))}{\mathbb{P}(H_0|\mathcal{D}(n))} \quad \text{(approximating the prior ratio by uniform prior)} \tag{6}$$

$$= s(n) \frac{\mathbb{P}(H_1|\mathcal{D}(n))}{1 - \mathbb{P}(H_1|\mathcal{D}(n))} \quad (H_0 \cup H_1 \text{ is the entire space}) \tag{7}$$

where $\mathbb{P}(H_1|\mathcal{D}(n)), \mathbb{P}(H_0|\mathcal{D}(n))$ are the corresponding posteriors. Note that, in the second line, we approximated the DP prior with a uniform prior over the existing answers.

When $n$ is sufficiently large compared with $\alpha$, $\mathbb{P}(H_1|\mathcal{D}(n))$ of the DP posterior can approximated by a Dirichlet distribution as:

$$\mathbb{P}(H_1|\mathcal{D}(n)) \approx \Pr[X_1 \geq \max_{i \neq 1} X_i, X \sim \text{Dirichlet}(N_1, N_2, \ldots, N_{s(n)}, \alpha)], \tag{8}$$

by approximating the probability of $A_1$ appearing in the base distribution $H$ to be zero. The Dirichlet distribution is a conjugate distribution of the categorical distribution of $s(n) + 1$ of answers, where the last dimension corresponds to the unobserved answers. Here, the final component of weight $\alpha$ is added to account for the base distribution $H$. While this quantity is not trivial to compute, it can be estimated using Monte Carlo methods by sampling from the Dirichlet distribution.

The following theorem states that, if we set $N_{\max}$ and $B$ sufficiently large, the performance of Algorithm 1 converges to the best-of-$\infty$ performance. The proof is given in Appendix C.

**Theorem 1.** (Consistency) *Assume that the LLM generates a finite number of answers $1, 2, \ldots, s$. For ease of discussion, let $p_j$ be the probability of answer $j$ and assume that $p_1 > p_2 \geq p_3 \geq \ldots \geq p_s > 0$. Namely, there are no ties for the most frequent answer, and each answer is generated with a non-zero probability. Then, as $N_{\max}, B \to \infty$, the algorithm's performance converges to the best-of-$\infty$ performance almost surely. Namely, the algorithm returns the true majority answer with probability $1$.*

## 3 LLM ENSEMBLE

Algorithm 1 is naturally extended to use more than one LLM. Let $i \in \mathcal{K}$ index the LLMs, and let $w = (w_1, w_2, \ldots, w_K)$ be the weight vector, where $w_i \geq 0$ and $\sum_{i \in \mathcal{K}} w_i = 1$. Algorithm 1 with an LLM ensemble proceeds as follows: for each generation, we first select an LLM $i$ with probability $w_i$, and then ask the selected LLM for the answer.

Let us consider the optimal weighting scheme for the BoN inference. Let $q \in \mathcal{Q}$ be the problem. Each problem is associated with answer domain $\mathcal{A}_q$.

---

[2]The use of the Bayes factor for categorical data is not new. Unlike their case, our case starts from an unknown number of categories, which is handled by the Dirichlet process prior and via some approximation. Aggarwal et al. (2023) applies Dirichlet distribution to majority voting in the context of LLM consistency, and approximated the posterior probability with Beta distribution on top-two majority answers. Wang et al. (2025a) applies frequentist confidence interval for adaptive stopping.

**Example 1** (AIME2025). *For AIME2025, $\mathcal{A}_q \subseteq \{0, 1, 2, \ldots, 999, U\}$, where $U$ denotes either an out-of-range integer, fractional number, or a failure to emit a final answer; $U$ is always incorrect.*

For each problem, let $g_q \in \mathcal{A}_q$ be the gold answer. Each LLM-problem pair $(i, q)$ is the probability distribution $\mathcal{P}_{iq}$ over $\mathcal{A}_q$. For each problem $q$, we obtain multiple generations from the LLMs and take a majority vote to produce $a_q$. The total number of correct answers is

$$f(\{a_q\}) := \sum_{q \in \mathcal{Q}} \mathbf{1}[a_q = g_q]. \tag{9}$$

We aim to maximize it in expectation: $\mathbb{E}[f(\{a_q\})]$. Here, the expectation is taken over the randomness in the generation of LLMs.

## 3.1 BEST-OF-ONE

Before going into Best-of-$\infty$, we first consider the best-of-one (Bo1) policy, which first selects an LLM with probability proportional to $w$, and then uses the LLM to generate a single answer. An immediate observation is that the optimal weight is to put all the weight on the best LLM.

**Lemma 1.** (Optimal Bo1) *The accuracy of Eq. equation 9 is maximized when we choose $w_{i^*} = 1$ and $w_j = 0$ for all $j \neq i^*$, where $w_{i^*}$ is the weight for the best LLM $i^*$. Namely, let $p_i^q = (p_{i,1}^q, p_{i,2}^q, \ldots, p_{i,|\mathcal{A}_q|}^q) \in \Delta_{\mathcal{A}_q}$ be the probability distribution on $\mathcal{A}_q$ of the answers that LLM $i$ generates. Then, $p_{i,g_q}^q$ be the probability that LLM $i$ generates the gold answer $g_q$ for problem $q$. The average accuracy of LLM $i$ is $\sum_q p_{i,g_q}^q$, and the best LLM, which maximizes this quantity, is $i^* = \arg\max_{i \in \mathcal{K}} \sum_q p_{i,g_q}^q$.*

*Proof.* It is easy to see that

$$f(\{a_q\}) := \sum_i w_i \left( \sum_{q \in \mathcal{Q}} p_{i,g_q}^q \right) \leq \max_i \left( \sum_{q \in \mathcal{Q}} p_{i,g_q}^q \right).$$

$\square$

For Bo1, the optimal weight is to put all the weight on the best LLM. However, this is no longer the case for BoN with $N > 1$. Put differently, under multi-generation majority voting, appropriately mixing non-optimal LLMs can be beneficial.

## 3.2 BEST-OF-$\infty$

As in the Bo1 setting, our design choice is to take a weighted majority vote with $w = (w_1, \ldots, w_K)$. When we consider the large-sample limit, the answer for problem $n$ is deterministic:[3]

$$a_q = \arg\max_j \left\{ \sum_{i \in \mathcal{K}} w_i p_{i,j} \right\},$$

with ties broken arbitrarily. Consequently $f(a_q)$ is also deterministic:

$$f(\{a_q\}) = \sum_{q \in \mathcal{Q}} \mathbf{1}[a_q = g_q].$$

Here, for ease of discussion, we omit the consideration for a tie. Henceforth, since our design choice is on the weight vector $w$, we denote it $f(w)$ and use $f(\{a_n\})$ and $f(w)$ interchangeably.

Our central question is how to choose a weight vector $w$ that maximizes the accuracy $f(w)$. The following lemma implies the hardness of optimizing $f(w)$.

**Lemma 2.** (Non-concavity) *$f(w)$ is a non-concave function on the simplex space of $w$.*

---

[3]We use the term deterministic to describe a non-random quantity.

*Proof.* Consider a dataset of just one question with two LLMs, where one LLM correctly answer the question and the other LLM fails. Namely, $f((1,0)) = 1$ and $f((0,1)) = 0$. Then the weighted combination is 0 at somewhere in between, which implies it is non-concave. □

While the proof above is an extremely simple case of two LLMs with a single problem, we will demonstrate the non-concavity in more complex cases.

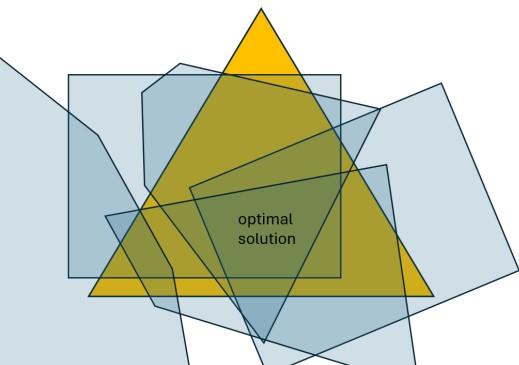

Figure 3: Visualization of the non-concave objective function $f(w)$ over the weight simplex $w$. The yellow simplex corresponds to $w$ in the simplex of the weights of the three LLMs. The gray region of the five polytopes (= five problems) are the region where the weighted majority of the corresponding weight correctly answer to the problem. The optimal solution is the intersection of four polytopes at the center, which corresponds to the case where four out of five problems are correctly answered.

Although non-concavity implies sub-optimality of gradient-based methods, a combinatorial optimization approach can be adopted for instances of typical scale. The crux in optimizing $f(w)$ is that the summand in equation 9 takes value one within a polytope.

**Lemma 3.** *(Polytope lemma) Let $\{p^q_{ij}\}_{i\in[K],j\in\mathcal{A}_q}$ be the arbitrary distributions of the answers. Then, the following set, which implies that answer $j$ is the most frequent answer, is a polytope:*

$$\left\{ w \in \Delta_K : \sum_i w_i p^q_{ij} > \max_{j' \neq j} \sum_i w_i p^q_{ij'} \right\}. \tag{10}$$

*Proof.* The region of equation 10 is an intersection of the following half-spaces:

$$w : \sum_i w_i p^q_{ij} > \sum_i w_i p^q_{ij'}$$

for all $j' \neq j$, which is a polyhedron. Since the desired space is an intersection of a polyhedron and a simplex $\Delta_{\mathcal{A}_q}$, it is finite. Therefore, it is a polytope. □

Lemma 3 states that the maximization on the number of correct answers is equivalent to the maximization on the number of polytopes that contain $w$ (Figure 3). By introducing auxiliary variable $y_q$ that indicates the correctness for each answer, this can be formulated as a mixed-integer linear programming (MILP) problem.

**Lemma 4.** *(MILP formulation) The equation 9 is equivalent to the following MILP problem:*

$$\max_{w \in \Delta^K, y \in \{0,1\}^N} \sum_q y_q \tag{11}$$

$$\text{s.t.} \quad w_i \geq 0 \; \forall_i \tag{12}$$

$$\sum_i w_i = 1 \tag{13}$$

$$A_q w \geq -m(1 - y_q) \; \forall q \tag{14}$$

*where $A_q$ is a matrix of size $\mathbb{R}^{|\mathcal{A}_q| \times K}$ such that its $j, i$ entry is $p^q_{i,g_q} - p^q_{i,j}$, and the $j$-th row corresponds to the fact that the total weight of the gold answer $g_q$ is larger than that of a wrong answer*

Table 1: Statistics of the large-scale generation dataset that we used in our experiments. Each file corresponds to a single answer. We release it at `https://figshare.com/s/ea10a6bd76bcf41e30bd`

| LLM | # of files | total generated tokens | total file size (MB) |
|---|---|---|---|
| AM-Thinking-v1 | 4,800 | 79,438,111 | 185.95 |
| Datarus-R1-14B-preview | 4,800 | 49,968,613 | 127.03 |
| EXAONE-Deep-32B | 60,640 | 478,575,594 | 1,372.35 |
| GPT-OSS-20B | 68,605 | 244,985,253 | 98.59[4] |
| LIMO-v2 | 6,095 | 77,460,567 | 219.45 |
| MetaStone-S1-32B | 60,757 | 806,737,009 | 2,458.48 |
| NVIDIA-Nemotron-Nano-9B-v2 | 60,640 | 295,466,626 | 897.82 |
| Phi-4-reasoning | 168,138 | 558,980,037 | 1,841.06 |
| Qwen3-4B | 20,640 | 547,170,887 | 1,704.28 |
| Qwen3-14B | 44,800 | 666,466,780 | 1,822.13 |
| Qwen3-30B-A3B-Thinking-2507 | 60,640 | 436,865,220 | 1,234.28 |

*$j$. The vector $m > 0$ is chosen sufficiently large, so that $A_q w \geq m$ is never satisfied when $A_q w$ has a negative component.*

The size of the problem instance depends on the number of LLMs $K$, the number of problems $N$, and the size of the possible set of answers $\mathcal{A}_q$. General MILP solving is NP-hard; in practice, however, open-source solvers scale smoothly to $K \approx 10^1$ LLMs and $N \approx 10^3$ problems, where typical size of $\mathcal{A}_q$ is $\approx 10^1$.

**Max margin solutions** As we illustrated in Figure 3, the objective function $f(w)$ has continuous region of optimal solutions. While any interior point on these position is optimal in best-of-$\infty$, its finite-$N$ performance can vary. In this paper, we adopt a "max margin" solution, that is at the most interior of the solution. Namely, we introduce a margin $\xi > 0$ and replaces $A_q w$ in equation 14 with $A_q w - \xi$. We choose the supremum of the margin $\xi$ such that the objective value $\sum_q y_q$ does not decrease, and adopts the solution on such margin. The optimization of margin can be done a binary search on the space of $\xi \in [0, m]$ where $m$ is a sufficiently large constant. This is a binary search problem of a monotone objective, which is practically feasible.

## 4 EXPERIMENTS

This section reports our experimental results. We considered heavy-reasoning tasks on open-weight LLMs that we can test on our local environment. We set Algorithm 1's hyperparameter $\alpha = 0.3$ for all the experiments. To solve MILPs, we use highspy, an open-source Python interface to the HiGHS optimization suite (Huangfu & Hall, 2018), which provides state-of-the-art solvers for large-scale LP, MIP, and MILP. We adopt the max-margin solution described in Section 3.2. Unless specified otherwise, all results are estimated from 100 independent runs. We use the Dirichlet posterior with each count $N_i(t) + 1$ rather than $N_i(t)$ to have faster stopping. The Bayes factor is calculated with 1,000 Monte Carlo samples from the posterior. Due to page limits, we show only several experimental results in the main text. More results are available in Appendix G.

### 4.1 TESTED OPEN-WEIGHT LLMS AND DATASETS

We evaluate open-weight LLMs ($\leq$ 32B parameters) across four reasoning benchmarks. We use the following problem sets: AIME2024 (Maxwell-Jia, 2024), AIME2025 (OpenCompass, 2025), GPQA-DIAMOND (Graduate-Level Google-Proof Q&A Benchmark; Rein et al. 2023), and MATH500 (Hendrycks et al., 2021). Details of the LLMs and datasets are provided in Appendix D. These datasets are challenging mathematical and scientific reasoning tasks. We did not test GSM8K (Cobbe et al., 2021) as it is too easy for the LLMs we tested.

---

[4]We do not use chain-of-thought (CoT) in our experiments and thus the file size is small; however, we also include an updated dataset that contains CoT.

**Large-scale generation dataset**  We generate a set of candidate answers by querying the LLM with the problem statement. For each pair of (LLM, problem), we generate at least 80 answers—an order of magnitude greater than the typical 8 generations reported in most LLM technical reports. We believe the difficulty of the problems as well as the scale of generated tokens are significantly larger than existing work on test-time computing.[5] Table 1 shows the statistics of the datasets used in our experiments. Base performance (Bo1, best-of-$\infty$) of these LLMs are shown in Appendix E. Every sample of answer in our subsequent experiments is drawn from this dataset. Best-of-$\infty$ performance is also estimated from these samples. We remove the unparseable answers, which benefits some of the LLMs with lower performance.

## 4.2 EXPERIMENTAL RESULTS

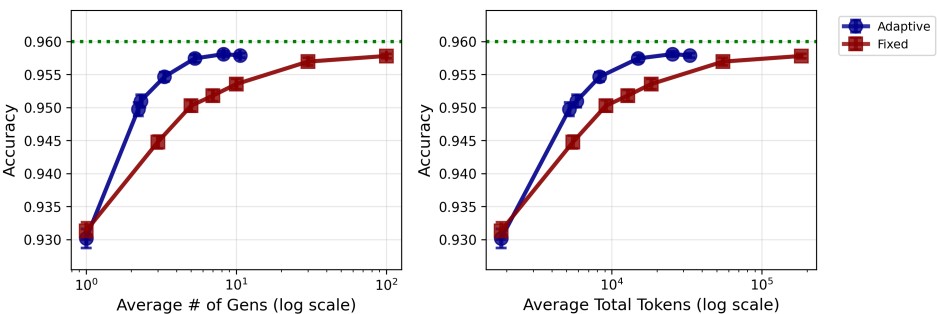

Figure 4: Cost-analysis of our proposed method and fixed BoN. GPT-OSS-20B on MATH500. "Adaptive" Algorithm 1 with average sample size of $\bar{N} = 3$ achieves the same accuracy as "fixed" sample of $N = 10$, and the algorithm with average sample size $\bar{N} \approx 10$ achieves the same accuracy as fixed $N = 100$. Thus, the adaptive sampling in this plot reduced the computation times by 2x-5x order. Both approach the best-of-$\infty$ performance (green dashed line).

**Experimental Set 1: Effectiveness of adaptive sampling**  First, we investigate the impact of adaptive sampling scheme of Algorithm 1 on the performance of majority voting. We set $N_{\max} = 100$ and tested varying Bayes factor $B = \{2, 3, 5, 7, 10, 30, 100, 300, \dots\}$. Figure 4 (left) compares the performance of Algorithm 1 with fixed budget of samples (BoN), where $x$-axis is the number of average samples per problem (log-scale), and $y$-axis is the accuracy. The figure clearly shows that the blue curve (Algorithm 1) achieves the same accuracy as the red curve (fixed BoN) with substantially fewer samples. Figure 4 (right) shows the average total number of tokens as a function of accuracy. The adaptive method again demonstrates a significant reduction in token usage to achieve the same accuracy level compared to the fixed method, although the gap is smaller than that of the sample count. This is because the adaptive method tends to stop sampling early for easier problems, which often require fewer tokens per generation.

**Experimental Set 2: Advantage of LLM ensemble over single LLM**  Second, we investigate the advantage of LLM ensemble over single LLM. We compare the performance of the single LLM with the optimal mixture of LLMs. The results in Figure 5 show that the ensemble method achieves higher accuracy than any single LLM, demonstrating the effectiveness of combining multiple models.

**Experimental Set 3: Learning a good weight**  Third, we investigate the generalization ability of our weight optimization method (Section 3). Figure 6 shows the performance of the learned weights as a function of the number of training problems on AIME2025. With five training problems, the learned weights approach the best single-LLM performance.

**Experimental Set 4: Transfer learning of the optimal weight**  To assess transferability, we trained weights on AIME2024 and tested on AIME2025; across 165 three-model combinations, the ensemble matched or exceeded the strongest individual model in 106 cases (64.2%).

---

[5]Also note that, for adaptive sampling scheme, around 80 samples are usually sufficient to achieve accuracy fairly close to the best-of-$\infty$ performance.

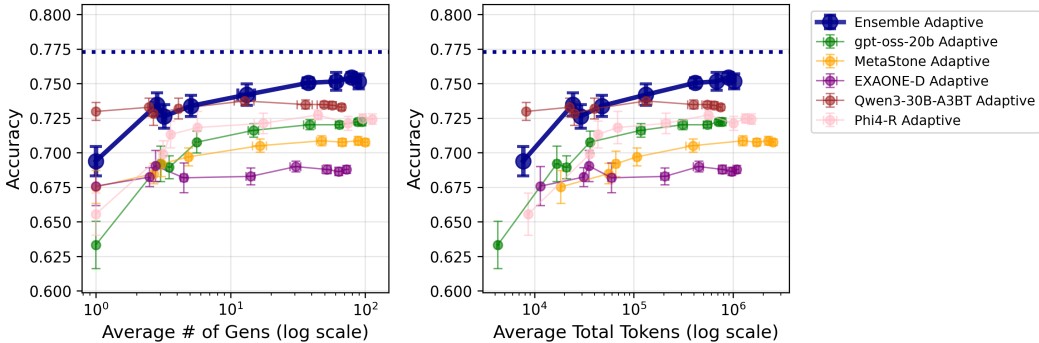

Figure 5: Performance comparison of the LLM ensemble of EXAONE-Deep-32B, MetaStone-S1-32B, Phi-4-reasoning, Qwen3-30B-A3B-Thinking, and GPT-OSS-20B on GPQA-Diamond. The weight is optimized to $w = (0.0176, 0.0346, 0.2690, 0.4145, 0.2644)$. The LLM ensemble outperforms any single LLM with $N \geq 5$ and approaches the blue dashed line of best-of-$\infty$ performance.

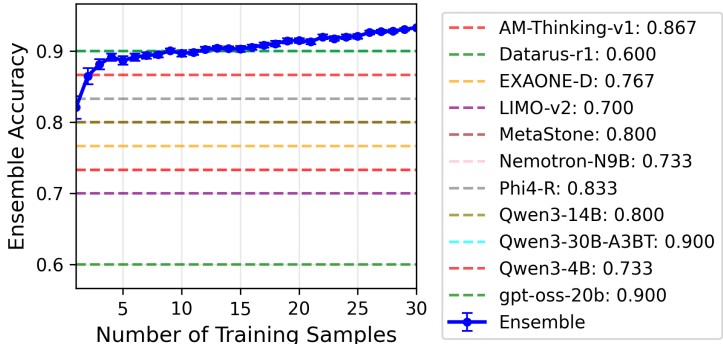

Figure 6: The number of samples to determine the weight (x-axis) as a performance of best-of-$\infty$ (y-axis) on AIME2025. The $x$-axis indicates the number of problems used to learn the weight and the $y$-axis indicates the best-of-$\infty$ performance with all problems. The score is averaged over 100 runs. The optimal weight has achieved the limit accuracy of 93.3%, whereas the best single LLM has the limit accuracy of 90.0%. Dashed lines indicate the best-of-$\infty$ performance of each LLM.

**Experimental Set 5: Comparison with other answer-selection methods** We finally compared the majority voting scheme with other selection scheme in the best-of-five (Bo5) test-time inference (Table 2). On AIME2025, majority voting outperforms random selection, self-certainty, reward models, and LLM-as-a-judge; full tables and settings are provided in the appendix (Appendix G.5).

## 5 CONCLUSION

In this paper, we view the majority voting as sampling from the underlying answer distribution, Its limit-$N$ performance naturally defined. To approximate this limit with a finite number of samples, we introduce an adaptive sampling method based on the Bayes factor. We also study the problem of aggregating responses from multiple LLMs and propose a majority voting that effectively leverages the strengths of individual models. The limit performance for large $N$ has advantage because the weights of LLM ensembles can be optimized by solving a mixed-integer linear programming problem. Through extensive experiments on challenging reasoning tasks, we have verified robustness across LLMs and problem sets. The scale of our generated answers is substantially larger than in existing work, and we release these generations for future research.

### ACKNOWLEDGMENTS

We thank Naoto Iwase for providing the guidance to use the MedRECT dataset.

Table 2: The accuracy of several selection methods on the best-of-five (Bo5) setting on the AIME2025 dataset. Answers are generated by GPT-OSS-20B. The scores are averaged over 16 trials and we report the two-sigma confidence intervals. Omniscient is a hypothetical upper bound that always selects the correct answer if it is present in the candidate answers, which requires the gold answer. Random, which selects one of $N$ answers uniformly at random, should match the performance of Bo1. Details of each method are described in Appendix G.5.

| Method | Mean $\pm$ CI |
| --- | --- |
| Omniscient | $91.04 \pm 1.32$ |
| Majority voting | $85.42 \pm 2.01$ |
| LLM-as-a-judge (tournament) | $82.92 \pm 2.57$ |
| LLM-as-a-judge (set) | $81.25 \pm 2.42$ |
| INF-ORM-Llama3.1-70B | $79.79 \pm 2.54$ |
| Skywork-Reward-V2-Llama-3.1-8B | $79.79 \pm 2.47$ |
| Skywork-Reward-V2-Qwen3-8B | $80.00 \pm 2.51$ |
| Self-certainty | $75.83 \pm 2.47$ |
| Random | $76.25 \pm 2.71$ |

## ETHICS STATEMENT

We acknowledge the potential ethical concerns surrounding the use of large language models (LLMs) in our research. Our work aims to improve the performance of LLMs in reasoning tasks, which may have implications for their deployment in real-world applications. Since we do not ask for LLM to work on any sensitive or harmful tasks, we believe that our work does not directly contribute to the generation of harmful content. We are committed to ensuring that our research is conducted responsibly and that the benefits of our work are shared broadly.

## REPRODUCIBILITY STATEMENT

We release the code and the generated answers used in our experiments to facilitate reproducibility. Detailed descriptions of the datasets, models, and experimental setups are provided in the Appendix.

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

## A  USAGE OF LLMS

We have used LLMs to write paragraphs as well as to find relevant literature. Programs are written with the aid of LLMs based on agentic programming tools (i.e., Cursor). However, all the responsibility of the content lies with us. Theorems and proofs are written by us.

## B  RELATED WORK

This section describes the related work on aggregating multiple answers from different individuals and LLMs.

### B.1  AGGREGATION OF MULTIPLE ANSWERS FROM INDIVIDUALS

**Controlling $N$ in majority voting**  Ensembling multiple predictions is a widely used technique for improving the accuracy of various machine learning tasks. One of the classic papers by Kolter & Maloof (2007) considers online ensemble learning, where the system can dynamically add or remove experts based on their performance. Regarding the optimal control of $N$, the idea closest to ours is the Urn model by Soto et al. (2016), which calculates the Bayesian probability that the empirical majority matches the true majority. However, their model samples without replacement, whereas LLM generation fits sampling with replacement. Their method also requires candidate answers and is thus not directly applicable to our setting. Motivated by ensembling methods for deep image classifiers, Inoue (2019) proposed an adaptive ensemble prediction method that adaptively aggregates the outputs of multiple probabilistic classifiers.

**Opinion aggregation in crowdsourcing**  A relevant lines of works in pre-LLM era is the opinion aggregation in crowdsourcing (Sheng et al., 2008; Li et al., 2017). One of the most popular methods in opinion aggregation is the method by David and Skene (Dawid & Skene, 1979). They introduced a probabilistic model that estimates the true labels of items by leveraging the agreement among multiple annotators. It comprises a confusion matrix $\pi$, whose $jk$ entry represents the probability such that each annotator's label is $j$ when the true label is $k$. Such a confusion matrix is not directly applicable to our setting because we cannot generally assume a fixed domain of answers in LLM generation. For example, in the AIME datasets, building a confusion matrix of 1,000 rows (possible answers are integers from 0 to 999) is not very practical. The Dawid-Skene model also assumes that the most frequent answer is the correct one. Subsequent works (Whitehill et al., 2009; Kajino et al., 2012; Takamatsu et al., 2012) addressed this issue by introducing a difficulty parameter for each problem. One of the largest difference between crowdsourcing and LLM ensemble is that the former typically assumes a single answer from each annotator, whereas the latter can generate multiple answers from the same LLM.

### B.2  AGGREGATION OF MULTIPLE ANSWERS FROM LLMS

Our method belong to a large umbrella of LLM Ensemble methods, where the forecaster uses multiple LLMs for a better output. A comprehensive survey on this topic (Chen et al., 2025b) categorizes ensemble LLM methods into several categories.[6] Our method falls into the category of "ensemble after inference", where we aggregate the outputs of multiple LLMs after they have generated their responses.

Within this category, Chen et al. (2025b) classified methods into three sub-categories: (1) selection, (2) selection-then-regeneration, and (3) cascade. The first directly selects the answer from generated outputs (our setting). The second selects a subset of LLMs and then merges their outputs using another LLM or a trained model. The third uses a cascade of LLMs, invoking a stronger model only when needed to save cost. Methods in (1) and (2) typically assume a fixed number of generations per LLM and optimize aggregation. In contrast, we primarily consider dynamically controlling the number of generations. Methods in (3) focus on minimizing total cost of calling LLMs.

---

[6]Figure 2 therein.

**(1) Selection** Li et al. (2024) proposed AgentForest that aggregates the predictions of multiple agents by using similarity agreement. Guha et al. (2024) introduced Smoothie, a graphical-model based method to choose the best LLM for each problem. Si et al. (2023) introduced Mixture of Reasoning Experts (MORE) framework that adopts multiple prompting strategy to obtain a mixture of experts and aggregates them by random forest classifier. Our methods belongs to this sub-category. Compared with these methods, our method is a simple average while others may use more complex aggregation strategies. Note also that these methods primarily consider a single generation per each LLM, whereas our paper primarily considers large number of generations per each LLM. A recent paper by Zhao et al. (2025a) proposes an aggregation method of multiple solutions by using reinforcement learning from verifiable rewards.

**(2) Selection-then-Regeneration** Jiang et al. (2023b) introduced LLM-Blender, an ensemble LLM method that comprises two modules: PAIRRANKER and GENFUSER. PairRanker chooses $K$ among $N$ LLMs, and GenFuser merges the outputs. Tekin et al. (2024) introduced LLM-TOPLA, an LLM ensemble method that maximizes the diversity of the answers. Based on the answer distribution of $N$ LLMs, they choose $K$ subset of LLMs that maximizes the diversity, and then train an aggregator (like multi-layer perceptron) that minimizes the cross-entropy loss. Lv et al. (2024) proposed an end-to-end method that integrates the subset selection and regeneration. Most of these methods are based on the idea of using many LLMs (or same LLM with different prompts) and single generation per each prompt, whereas our paper primarily considers a relatively small subset of LLMs for each prompt, and large number of generations per each LLM.

**(3) Cascading** Varshney & Baral (2022) is one of the earliest work that introduced cascading. Yue et al. (2024) proposed an aggregation of weak and strong LLMs. In their model, if the weak LLM and the cascade LLM disagree with the answer, then the strong LLM is invoked. The primal motivation in cascading is to save the cost of calling strong LLMs, which is orthogonal to our goal of improving the accuracy of maximizing the accuracy given large amount of computation.

**Answer selection based on reward models and LLM-as-a-judge** A common approach to aggregate multiple answers from LLMs is to use reward models or LLM-as-a-judge methods. Typically, reward models are constructed on top of language models. These approaches can be broadly categorized into two groups: those in which the reward model directly outputs a scalar value (Rafailov et al., 2023; Liu et al., 2024), and those in which the reward model provides comparative judgments or rankings over multiple responses (Mahan et al., 2024; Dong et al., 2024; Son et al., 2024; Guo et al., 2025; Chen et al., 2025a). The methods of the latter category are referred to as generative reward models, reward reasoning models, or LLM-as-a-judge. Compared to our approach, these methods incur additional computational cost due to the reliance on reward models. Also, in our experiments, we did not observe particular advantage of using reward models (see Table 2).

## C  PROOF OF THEOREM 1

*Proof of Theorem 1.* Let $\hat{p}_a(n) = N_j(n)/n$ be the empirical mean of answer $j$ at round $n$. Hoeffding's inequality implies that

$$\mathbb{P}[|\hat{p}_a(n) - p_a| \geq \epsilon] \leq 2\exp(-2n\epsilon^2).$$

Let $\Delta = \min_{j \neq 1}(p_1 - p_j) > 0$ be the gap between the most frequent answer and the second most frequent answer. Then it holds that

$$\mathbb{P}\left[\bigcap_{n=N_0}^{\infty} |\hat{p}_j(n) - p_j| \geq \frac{\Delta}{2}\right] \leq \sum_{n=N_0}^{\infty} \mathbb{P}\left[|\hat{p}_j(n) - p_j| \geq \frac{\Delta}{2}\right] \qquad \text{(Union bound)}$$

$$\leq \sum_{n=N_0}^{\infty} 2\exp\left(-n\frac{\Delta^2}{2}\right) \qquad \text{(Hoeffding's inequality)}$$

$$= \frac{2e^{-N_0\Delta^2/2}}{1 - e^{-\Delta^2/2}}. \qquad (15)$$

and by choosing $N_0 = N_0(\delta)$ sufficiently large, the right-hand side can be made no larger than $\delta/s$. Union bound over all $s$ answers implies that, with probability at least $1 - \delta$, it holds that

$$\hat{p}_1(n) - \hat{p}_j(n) \geq p_1 - p_j - 2 \times \frac{\Delta}{2} \geq 0, \forall j \neq 1, \forall n \geq N_0(\delta).$$

Namely, at least with probability $1-\delta$, the empirical most frequent answer is indeed the true majority answer for all $n \geq N_0(\delta)$, and thus, if stopping time is longer than $N_0(\delta)$, the algorithm returns the true majority answer. By choosing $N_{\max} \geq N_0(\delta)$ and $B$ sufficiently large[7], the algorithm stops after $N_0(\delta)$ with probability 1, and thus, the algorithm returns the true majority answer with probability at least $1 - \delta$. Since $\delta > 0$ is arbitrary, the algorithm returns the true majority answer with probability arbitrarily close to 1. Proof of Theorem 1 is complete. □

**Remark 1.** (Frequentist stopping criteria) *While Dirichlet posterior naturally fits with our task, we may consider frequentist stopping criteria based on the observed data. Advantages of the frequentist approach include its closed formula as well as rigorous guarantee in view of a frequentist. A drawback is that its configuration of the hyperparameter tends to be conservative: the confidence level that it requires is often higher than what actually is, potentially leading to oversampling. To bound the error probability, it needs to consider the correction due to adaptive sampling (Kaufmann & Koolen, 2021), as well as a multiple-testing correction with respect to the size of answer set $s(t)$. The latter seems particularly problematic, as $s(t)$ is unknown and potentially unbounded. For this reason, we do not see any existing work that adopts a frequentist approach to testing adaptive majority voting. For example, existing methods on majority voting, such as Soto et al. (2016), which we will elaborate in Section B.1, also adopt Bayesian approach. Therefore, we do not pursue this direction in this paper.*

## C.1 FINITE-TIME ANALYSIS OF STOPPING TIME

In this section, we conduct a finite-time analysis of the stopping time of Algorithm 1.

**Theorem 2.** (Finite-time stopping) *Algorithm 1 stops within*

$$O\left(\frac{1}{\Delta^2} \log\left(|\mathcal{A}| \max(B, 1/\delta)\right)\right)$$

*rounds with probability at least $1 - \delta$, where $\Delta$ is the gap between the most frequent answer and the second most frequent answer, and $\mathcal{A}$ is the set of possible answers by LLM.*

Note that this rate is optimal for $\delta$-correct identification because of the lower bound of the best-arm identification problem (e.g., Garivier & Kaufmann (2016)) with two arms, which is about identifying the larger of two Bernoulli distributions with gap $\Delta$, is $\Omega(\log(\delta^{-1})/\Delta^2)$ as well.

*Proof of Theorem 2.* Since we focus on a particular problem $q$ we drop $q$ and denote $a_g$ be the gold answer and $\mathcal{A}$ be the set of possible answers by LLM.

Let $P(n) = \Pr[X_1 \geq \max_{i \neq 1} X_i, X \sim \text{Dirichlet}(N_1+1, N_2+1, \dots, N_{s(n)}+1, \alpha)]$. Let $P_{a_g, a}(n)$ be the Beta posterior probability such that the parameter of answer $a_g$ is larger than that of answer $a$. By using the fact that a Dirichlet distribution restricted to two dimensions is a Beta distribution, it is equivalent to

$$1 - \mathbb{P}[X \geq 1/2], X \sim \text{Beta}(N_a, N_{a_g}). \tag{16}$$

---

[7]This is because, the possible combination of answers with the first $N_0$ samples is finite, and thus, the possible value[8] of BF that it can take until the first $N_0$ sample is finite. If we set $B$ larger than that the largest of such values, then the algorithm never stops before the $N_0$ samples.

A sufficient condition for stopping at round $n$ is (c.f., Eq. equation 7 and Eq. equation 8) is

$$\left\{B \geq g(n)\frac{P(n)}{1 - P(n))}\right\} \supseteq \left\{B \geq \frac{P(n)}{1 - P(n))}\right\} \tag{17}$$

$$\supseteq \left\{2B \geq \frac{1}{1 - P(n))}\right\} \quad \text{(for } B \geq 2\text{)} \tag{18}$$

$$\supseteq \left\{P(n) \geq 1 - \frac{1}{2B}\right\} \tag{19}$$

$$\supseteq \bigcap_{a \neq a_g} \left\{P_{a_g,a}(n) \geq 1 - \frac{1}{2|\mathcal{A}|B}\right\}. \tag{20}$$

By Hoeffding's inequality, for any $a$,

$$|\hat{p}_a(n) - p_a| \leq \frac{\Delta}{4} \tag{21}$$

holds with probability at least $1 - \exp(-n\Delta^2/8)$. If we fix $n$ such that

$$n \geq \frac{8}{\Delta^2}\log\left(\frac{|\mathcal{A}|}{\delta}\right), \tag{22}$$

then equation 21 holds for all $a \in \mathcal{A}$ with probability at least $1 - \delta$. Under equation 21, it holds that

$$\hat{p}_{a_g}(n) - \hat{p}_a(n) \geq p_{a_g} - p_a - 2 \times \frac{\Delta}{4} \geq \Delta - \frac{\Delta}{2} = \frac{\Delta}{2}. \tag{23}$$

for all $a \neq a_g$. Therefore, by letting $\mu = \hat{p}_a(n)/(\hat{p}_{a_g}(n) + \hat{p}_a(n)) < 1/2$, we have

$$P_{a_g,a}(n) \geq 1 - \frac{7}{1/2 - \mu}\exp(-nd(\mu, 1/2)) \quad \text{(by Lemma 5)} \tag{24}$$

If we choose $n$ such that

$$n \geq \frac{1}{d(\mu, \frac{1}{2})}\log\left(\frac{7|\mathcal{A}|B}{\frac{1}{2} - \mu}\right), \tag{25}$$

then

$$1 - \frac{7}{1/2 - \mu}\exp(-nd(\mu, 1/2)) \geq 1 - \frac{1}{2|\mathcal{A}|B}. \tag{26}$$

In summary, if we choose $n$ such that both equation 22 and equation 25 hold, then the algorithm stops at (or before) $n$ with probability at least $1 - \delta$. Regarding the order of equation 25, we have

$$n = \frac{1}{d(\mu, \frac{1}{2})}\log\left(\frac{7|\mathcal{A}|B}{\frac{1}{2} - \mu}\right) \tag{27}$$

$$= O\left(\frac{1}{\Delta^2}\log\left(|\mathcal{A}|B\right)\right) \quad \text{(Pinsker's inequality: } d(p,q) \geq 2(p-q)^2\text{)} \tag{28}$$

$$\tag{29}$$

and thus, $n$ such that both equation 22 and equation 25 hold is

$$n = O\left(\frac{1}{\Delta^2}\log\left(|\mathcal{A}|\max(B, 1/\delta)\right)\right). \tag{30}$$

$$\square$$

**Lemma 5** (Beta tail). *Let* $X \sim \text{Beta}(1 + n\mu, 1 + n(1 - \mu))$ *be a random variable following the Beta distribution. Then, for any* $a > \mu$, *it holds that*

$$\mathbb{P}[X \geq a] \leq \frac{7}{a - \mu}\exp(-nd(\mu, a)).$$

*where* $d(\mu, a) = \mu\log\frac{\mu}{a} + (1 - \mu)\log\frac{1-\mu}{1-a}$ *is the KL divergence between two Bernoulli distributions.*

*Proof of Lemma 5.* Let $B(a,b)$ and $\Gamma(x)$ be the Beta function and the gamma function, respectively.

$$\mathbb{P}[X \geq a] \tag{31}$$

$$= \frac{1}{B(1+n\mu, 1+n(1-\mu))} \int_a^1 x^{n\mu}(1-x)^{n(1-\mu)}dx \tag{32}$$

$$= \frac{1}{B(1+n\mu, 1+n(1-\mu))} \left( \left[ \frac{1}{\frac{n\mu}{x} - \frac{n(1-\mu)}{(1-x)}} \cdot x^{n\mu}(1-x)^{n(1-\mu)} \right]_a^1 - \int_a^1 \frac{n\mu + n(1-\mu)}{(\frac{n\mu}{x} - \frac{n(1-\mu)}{(1-x)})^2} x^{n\mu}(1-x)^{n(1-\mu)}dx \right) \tag{33}$$

(by integration by parts) $\tag{34}$

$$\leq \frac{1}{B(1+n\mu, 1+n(1-\mu))} \frac{1}{\frac{n(1-\mu)}{(1-1)}\frac{n\mu}{a}} a^{n\mu}(1-a)^{n(1-\mu)} \tag{35}$$

$$\leq \frac{\Gamma(2+n)}{\Gamma(1+n\mu)\Gamma(1+n(1-\mu))} \frac{a(1-a)}{n(a-\mu)} a^{n\mu}(1-a)^{n(1-\mu)} \tag{36}$$

$$\tag{37}$$

and by Stirling's formula $\sqrt{2\pi} \leq \frac{\Gamma(z)}{z^{z-1/2}e^{-z}} \leq \sqrt{2\pi}e^{1/12}$, we have

$$\mathbb{P}[X \geq a] \leq \frac{a(1-a)e^{1/12}}{n(a-\mu)\sqrt{2\pi}} \sqrt{\frac{(n+2)^3}{(n\mu+1)(n(1-\mu)+1)}} \frac{(n+2)^n}{(n\mu)^{n\mu}(n(1-\mu))^{n(1-\mu)}} a^{n\mu}(1-a)^{n(1-\mu)} \tag{38}$$

$$\leq \frac{a(1-a)e^{1/12}}{n(a-\mu)\sqrt{2\pi}} \sqrt{\frac{27n^3}{n(1-\mu)}} e^2 e^{-nd(\mu,a)} \tag{39}$$

$$\leq \frac{ae^{25/12}}{(a-\mu)\sqrt{2\pi}} \sqrt{\frac{27}{1-a}} \exp(-nd(\mu,a)) \tag{40}$$

$$\leq \frac{e^{25/12}}{5(a-\mu)} \sqrt{\frac{54}{\pi}} e^{-nd(\mu,a)} \tag{41}$$

$$\leq \frac{7}{a-\mu} e^{-nd(\mu,a)} \tag{42}$$

and the proof is complete. $\qquad\square$

# D  LIST OF LLMS AND PROBLEM SETS

We tested the following LLMs. The model temperature is 0.6 unless otherwise specified. We follow the model recommendation to set the temperature and other hyperparameters. The maximum model length is $\min(X, \text{maximum context length of LLM}) - 2500$ tokens, where $X = 100000$ all but GPQA-DIAMOND, whereas $X = 50000$ for GPQA-DIAMOND. The 2500 token margin is reserved for the prompt; we believe that this does not matter to MATH500 and GPQA-DIAMOND at all, and to AIME2024/2025 very slightly.

- Phi-4-reasoning (Abdin et al., 2025) is a 14-billion-parameter (14B) reasoning-oriented model developed by Microsoft, released in April 2025. It builds on the Phi-4 base model using supervised fine-tuning on a dataset of chain-of-thought traces and reinforcement learning. We set temperature to 0.8.
- GPT-OSS-20B (OpenAI, 2025) is the smaller version of the two LLMs released in October 2025 by OpenAI. This model has 21B parameters in total. We set the reasoning effort to be medium (default setting).
- AM-Thinking-v1 (Ji et al., 2025) is a 32B dense model released in May 2025 by the a-m-team. It is built upon the pre-trained Qwen 2.5-32B-Base, then enhanced through a specialized post-training pipeline featuring Supervised Fine-Tuning (SFT) followed by reinforcement learning (RL).

- EXAONE-Deep-32B (LG AI Research, 2025) is a 32B model released in May 2025 by LG AI Research as part of the EXAONE Deep series. Built with 64 Transformer layers, a 102K vocabulary, and a 32K-token context window, it is designed to excel in reasoning-intensive tasks such as mathematics and coding.

- Nemotron-Nano-9B (NVIDIA, 2025) is a 9-billion-parameter hybrid reasoning model by NVIDIA, released in August 2025. It features a Mamba-2 + Transformer hybrid architecture, replacing most attention layers with efficient Mamba-2 layers. It was pretrained from scratch (using a 12B base model over 20 trillion tokens) and then compressed via distillation. Post-training includes SFT, GRPO, DPO, and RLHF.

- MetaStone-S1-32B (Wang et al., 2025b) is a 32B reflective generative reasoning model, released around July 2025. It introduces a novel Reflective Generative Form, merging policy generation and process reward modeling within a single shared backbone, enabled by a lightweight Self-supervised Process Reward Model (SPRM).

- Qwen3, released in April 2025 by Alibaba Cloud (Qwen Team, 2025), is the third-generation open-source large language model family featuring hybrid reasoning, long context support, agentic capabilities, and multilingual fluency. We use three versions of Qwen3. Namely, Qwen3-4B, Qwen3-14B, and Qwen3-30B-A3B-Thinking-2507.

- LIMO-v2 (Ye et al., 2025) is a 32B Qwen2.5-based reasoning model released in July 2025, fine-tuned on ∼800 carefully curated samples to achieve top-tier math reasoning with remarkable data efficiency—embodying the "Less-Is-More" principle.

We tested the following datasets:

- AIME2024 (Maxwell-Jia, 2024) consists of 30 problems that were used American Invitational Mathematics Examination (AIME) held during January 31 and February 1, 2024. AIME2024 tests mathmatical problem-solving skills in vast field of mathmatical topics. High-scoring high-school students are invited to participate in the United States of America Mathematics Olympiad (USAMO). All answers are integers between 1–999.

- AIME2025 (OpenCompass, 2025) consists of 30 problems that were used American Invitational Mathematics Examination (AIME) held from February 10 to February 12, 2025. Its format is identical to AIME2024.

- GPQA-DIAMOND (Graduate-Level Google-Proof Q&A Benchmark, Rein et al. 2023) is a set of multiple-choice questions crafted by PhD-level experts in biology, physics, and chemistry. The Diamond is a subset of 198 GPQA problems that distinguishes Ph.D. level experts from the others. The answers are in multiple-choice format (A–D).

- MATH500 (Hendrycks et al., 2021) is a benchmark derived from the MATH dataset, which contains challenging competition-level mathematics problems covering algebra, geometry, number theory, probability, and other advanced topics. The MATH500 subset consists of 500 carefully selected problems used in recent evaluation studies, and is designed to test mathematical problem-solving skills beyond high-school level. All problems require generating detailed reasoning and solutions rather than multiple-choice responses. The answer format varies, including numeric integers, fractions, complex numbers, and vectors.

Among these datasets, AIME2024/2025 benefits for a long chain of thought (CoT) reasoning, as the problems are challenging and require multi-step reasoning.[9] GPQA-DIAMOND and MATH500 also require long CoT, but the benefit of it is less significant than AIME2024/2025. We did not include GSM8K (Cobbe et al., 2021) because these problems are relatively easy and finishes with a short CoT for the tested LLMs, and thus the benefit of ensemble was not significant.

---

[9]Regarding the scaling of performance as a function of CoT length, see, e.g., Figure 1 of Muennighoff et al. (2025) and Figure 7 of Yan et al. (2025).

Table 3: Summary performance per model across datasets. The scores are estimated from at least 80 generation for each model and dataset. GPQA-D is an abbreviation of GPQA-DIAMOND.

| LLM | AIME2024 | | AIME2025 | | GPQA-D | | MATH500 | |
|---|---|---|---|---|---|---|---|---|
| | Bo1 | Bo∞ | Bo1 | Bo∞ | Bo1 | Bo∞ | Bo1 | Bo∞ |
| AM-Thinking-v1 | 0.789 | 0.900 | 0.762 | 0.867 | – | – | – | – |
| Datarus-R1-14B-preview | 0.516 | 0.733 | 0.370 | 0.600 | – | – | – | – |
| EXAONE-Deep-32B | 0.715 | 0.867 | 0.627 | 0.767 | 0.661 | 0.692 | 0.945 | 0.962 |
| GPT-OSS-20B | 0.780 | 0.900 | 0.744 | 0.900 | 0.642 | 0.722 | 0.928 | 0.960 |
| LIMO-v2 | 0.620 | 0.800 | 0.527 | 0.700 | – | – | – | – |
| MetaStone-S1-32B | 0.820 | 0.867 | 0.747 | 0.800 | 0.670 | 0.707 | 0.947 | 0.950 |
| NVIDIA-Nemotron-Nano-9B-v2 | 0.716 | 0.867 | 0.600 | 0.733 | 0.584 | 0.626 | 0.938 | 0.956 |
| Phi-4-reasoning | 0.729 | 0.867 | 0.643 | 0.833 | 0.658 | 0.727 | 0.878 | 0.944 |
| Qwen3-4B | 0.735 | 0.800 | 0.655 | 0.733 | – | – | – | – |
| Qwen3-14B | 0.830 | 0.867 | 0.744 | 0.800 | – | – | 0.946 | 0.956 |
| Qwen3-30B-A3B-Thinking-2507 | 0.905 | 0.933 | 0.858 | 0.900 | 0.720 | 0.732 | 0.954 | 0.960 |

# E   Bo1 AND BEST-OF-∞ PERFORMANCE OF EACH MODEL

We list Bo1 (averaged) and best-of-∞ performance of each model in Table 3. We have used the same prompt (Section E.1) for all models, which might be sub-optimal for some models. The evaluated performance also depends on the answer parser. While we used the consistent and a reasonably flexible answer parser for all models, we acknowledge some examples[10] where the parsing is imperfect. We have not specified any tool call option. Also note that GPT-OSS-20B's reasoning mode is set to medium (default setting), which is the second best setting. Finally, we clarify our goal is not to argue superiority of some models over the others, but to give some idea on the performance of each model that we use for the verification of our methods of adaptive sampling (Algorithm 1).

## E.1   PROMPTS FOR ANSWER GENERATION

We send the following request to a LLM that we launched as a vllm process:

{"role": "user", "content": prompt}

where the examples of the prompt are given below: The first prompt is from AIME2024, and the second prompt is from GPQA-DIAMOND.

```
Let $x,y$ and $z$ be positive real numbers that satisfy the
    following system of equations:
\[\log_2\left({x \over yz}\right) = {1 \over 2}\]
\[\log_2\left({y \over xz}\right) = {1 \over 3}\]
\[\log_2\left({z \over xy}\right) = {1 \over 4}\]
Then the value of $\left|\log_2(x^4y^3z^2)\right|$ is $\tfrac{m}{n}$
    where $m$ and $n$ are relatively prime positive integers. Find
    $m+n$.
 Please reason step by step, and put your final answer within \boxed
    {}.
```

```
Among the following exoplanets, which one has the highest density?

a) An Earth-mass and Earth-radius planet.
b) A planet with 2 Earth masses and a density of approximately 5.5 g
    /cm^3.
```

---

[10]In particular, MATH500 where the answer format varies.

```
c) A planet with the same composition as Earth but 5 times more
    massive than Earth.
d) A planet with the same composition as Earth but half the mass of
    Earth.

A. d
B. a
C. b
D. c
 Please reason step by step, and put your final answer as the letter
     choice (A), (B), (C), etc. within \boxed{}.
```

For NVIDIA Nemotron-Nano-9B, we prepend the recommended system message "/think".

## E.2 PROMPTS FOR LLM-AS-A-JUDGE

The following illustrates a prompt used to instruct an LLM-as-a-judge to select the best answer among a set of candidates. In this prompt, last_part_1, last_part_2, ... denote the final 5000 characters of each answer preceding the </think> tag.

```
Please evaluate the following 5 answer excerpts for this
    mathematical problem and determine which answer you think is the
     most correct.

Problem:
Let $x,y$ and $z$ be positive real numbers that satisfy the
    following system of equations:
\[\log_2\left({x \over yz}\right) = {1 \over 2}\]
\[\log_2\left({y \over xz}\right) = {1 \over 3}\]
\[\log_2\left({z \over xy}\right) = {1 \over 4}\]
Then the value of $\left|\log_2(x^4y^3z^2)\right|$ is $\tfrac{m}{n}$
     where $m$ and $n$ are relatively prime positive integers. Find
    $m+n$.

Answer 1 (Last 5000 chars before </think>):
{last_part_1}

Answer 2 (Last 5000 chars before </think>):
{last_part_2}

Answer 3 (Last 5000 chars before </think>):
{last_part_3}

Answer 4 (Last 5000 chars before </think>):
{last_part_4}

Answer 5 (Last 5000 chars before </think>):
{last_part_5}

Among the above 5 answer excerpts (showing the last parts before </
    think> tag), which answer do you think is the most correct,
    logical, and complete?

Please provide detailed reasoning for your judgment, and then output
     the number of the answer you think is correct (1, 2, 3, 4, 5)
    enclosed in \boxed{}.

Example: \boxed{1}

Judgment:
```

Table 4: Basic performance for each problem. The final line at column "accuracy" indicates Bo1 performance, and the final line at "majority answer" indicates best-of-∞ performance. LLM=Phi-4-reasoning, Dataset=AIME2025.

| Problem No. | Total answers | Correct answers | Accuracy | Gold answer | Majority answer |
|---|---|---|---|---|---|
| 1 | 160 | 159 | 0.994 | 70 | 70 |
| 2 | 160 | 112 | 0.700 | 588 | 588 |
| 3 | 160 | 154 | 0.963 | 16 | 16 |
| 4 | 160 | 150 | 0.938 | 117 | 117 |
| 5 | 160 | 146 | 0.912 | 279 | 279 |
| 6 | 160 | 158 | 0.988 | 504 | 504 |
| 7 | 160 | 96 | 0.600 | 821 | 821 |
| 8 | 160 | 147 | 0.919 | 77 | 77 |
| 9 | 160 | 134 | 0.838 | 62 | 62 |
| 10 | 160 | 58 | 0.362 | 81 | 81 |
| 11 | 160 | 120 | 0.750 | 259 | 259 |
| 12 | 160 | 137 | 0.856 | 510 | 510 |
| 13 | 160 | 5 | 0.031 | 204 | 487/3 |
| 14 | 160 | 5 | 0.031 | 60 | 63 |
| 15 | 160 | 0 | 0.000 | 735 | 147 |
| 16 | 160 | 158 | 0.988 | 468 | 468 |
| 17 | 160 | 157 | 0.981 | 49 | 49 |
| 18 | 160 | 87 | 0.544 | 82 | 82 |
| 19 | 160 | 154 | 0.963 | 106 | 106 |
| 20 | 160 | 114 | 0.713 | 336 | 336 |
| 21 | 160 | 143 | 0.894 | 293 | 293 |
| 22 | 160 | 45 | 0.281 | 237 | 60671 |
| 23 | 160 | 66 | 0.412 | 610 | 610 |
| 24 | 160 | 77 | 0.481 | 149 | 149 |
| 25 | 160 | 132 | 0.825 | 907 | 907 |
| 26 | 160 | 111 | 0.694 | 113 | 113 |
| 27 | 160 | 136 | 0.850 | 19 | 19 |
| 28 | 160 | 1 | 0.006 | 248 | 625 |
| 29 | 160 | 75 | 0.469 | 104 | 104 |
| 30 | 160 | 48 | 0.300 | 240 | 240 |
| total | 4800 | 3085 | 0.643 | | 0.833 |

### E.3 SOURCE CODE

Our source code is available at https://figshare.com/s/8bd1830a255278e57830.

## F COMPLEMENTARITY IN LLM ENSEMBLES FOR AIME 2025

In the AIME2025 dataset, we explored the combination of Phi-4-reasoning (Table 4) and GPT-OSS-20B (Table 5) to enhance performance on complex reasoning tasks. By leveraging the strengths of both models, we aimed to achieve better accuracy and robustness in our predictions. In this case, Phi-4-reasoning can solve Problem 30 that GPT-OSS-20B cannot solve, and can complement the performance. As a result, its LLM ensemble achieved 0.933 best-of-∞ accuracy, which is higher than the individual accuracies of Phi-4-reasoning (0.733) and GPT-OSS-20B (0.900). This demonstrates the effectiveness of combining different models to improve overall performance on challenging tasks.

Table 5: Basic performance for each problem. The final line at column "accuracy" indicates Bo1 performance, and the final line at "majority answer" indicates best-of-$\infty$ (limit) performance. LLM=GPT-OSS-20B, Dataset=AIME2025.

| Problem No. | Total answers | Correct answers | Accuracy | Gold answer | Majority answer |
|---|---|---|---|---|---|
| 1 | 85 | 85 | 1.000 | 70 | 70 |
| 2 | 85 | 76 | 0.894 | 588 | 588 |
| 3 | 85 | 85 | 1.000 | 16 | 16 |
| 4 | 85 | 83 | 0.976 | 117 | 117 |
| 5 | 85 | 81 | 0.953 | 279 | 279 |
| 6 | 85 | 85 | 1.000 | 504 | 504 |
| 7 | 85 | 52 | 0.612 | 821 | 821 |
| 8 | 85 | 80 | 0.941 | 77 | 77 |
| 9 | 85 | 75 | 0.882 | 62 | 62 |
| 10 | 85 | 53 | 0.624 | 81 | 81 |
| 11 | 85 | 61 | 0.718 | 259 | 259 |
| 12 | 85 | 56 | 0.659 | 510 | 510 |
| 13 | 85 | 17 | 0.200 | 204 | 204 |
| 14 | 85 | 3 | 0.035 | 60 | 74 |
| 15 | 85 | 0 | 0.000 | 735 | 147 |
| 16 | 85 | 81 | 0.953 | 468 | 468 |
| 17 | 85 | 85 | 1.000 | 49 | 49 |
| 18 | 85 | 62 | 0.729 | 82 | 82 |
| 19 | 85 | 84 | 0.988 | 106 | 106 |
| 20 | 85 | 79 | 0.929 | 336 | 336 |
| 21 | 85 | 72 | 0.847 | 293 | 293 |
| 22 | 85 | 85 | 1.000 | 237 | 237 |
| 23 | 85 | 45 | 0.529 | 610 | 610 |
| 24 | 85 | 58 | 0.682 | 149 | 149 |
| 25 | 85 | 81 | 0.953 | 907 | 907 |
| 26 | 85 | 72 | 0.847 | 113 | 113 |
| 27 | 85 | 81 | 0.953 | 19 | 19 |
| 28 | 85 | 36 | 0.424 | 248 | 248 |
| 29 | 85 | 71 | 0.835 | 104 | 104 |
| 30 | 85 | 14 | 0.165 | 240 | 188 |
| total | 2550 | 1898 | 0.744 | | 0.900 |

# G  ADDITIONAL EXPERIMENTS

To verify the robustness of our findings, we conducted similar experiments on other LLMs and datasets. The results are consistent with the main experiments in the paper, confirming the robustness of our proposed methods across different settings. As is the main paper, all error bars are standard two-sigma confidence intervals.

## G.1  EXPERIMENTAL SET 1: EFFECTIVENESS OF ADAPTIVE SAMPLING

In the following pages, we present the performance comparison between our proposed adaptive algorithm (Algorithm 1) and the fixed-sample BoN across various LLMs and datasets (Figures 8–11). The results consistently demonstrate that our adaptive approach outperforms the fixed-sample-size method given the same number of generation (= samples) or the same token budget. This is because our algorithm is adaptive; for easy problems where the model always outputs the same answer, it uses fewer samples, while for hard problems where the model's answers vary, it uses more samples. This adaptivity leads to better overall performance compared to a fixed-sample-size approach.

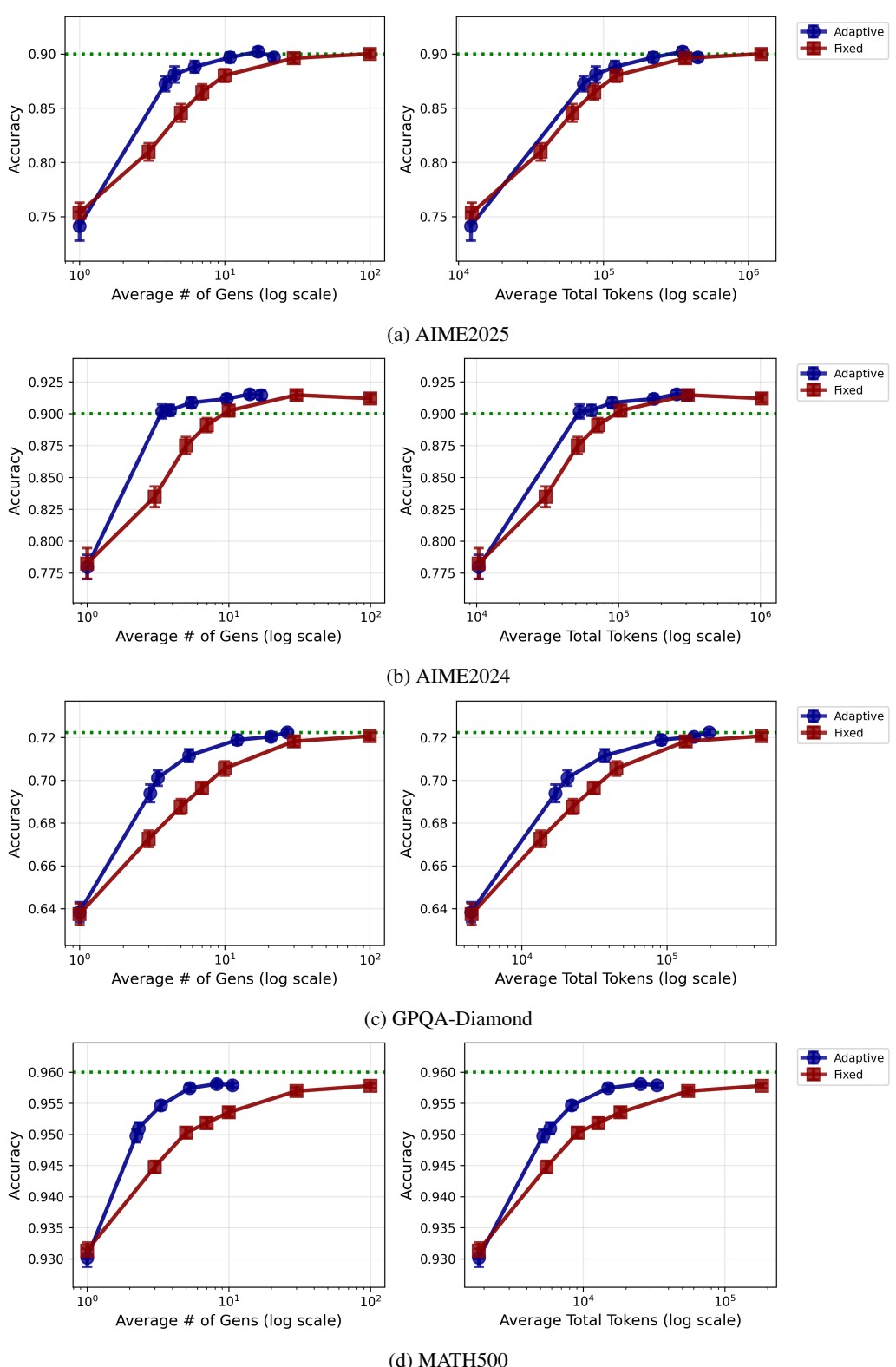

Figure 7: Cost-analysis of our proposed method and fixed BoN for GPT-OSS-20B. The error bars are standard two-sigma confidence intervals. Green dashed line indicates the best-of-$\infty$ performance.

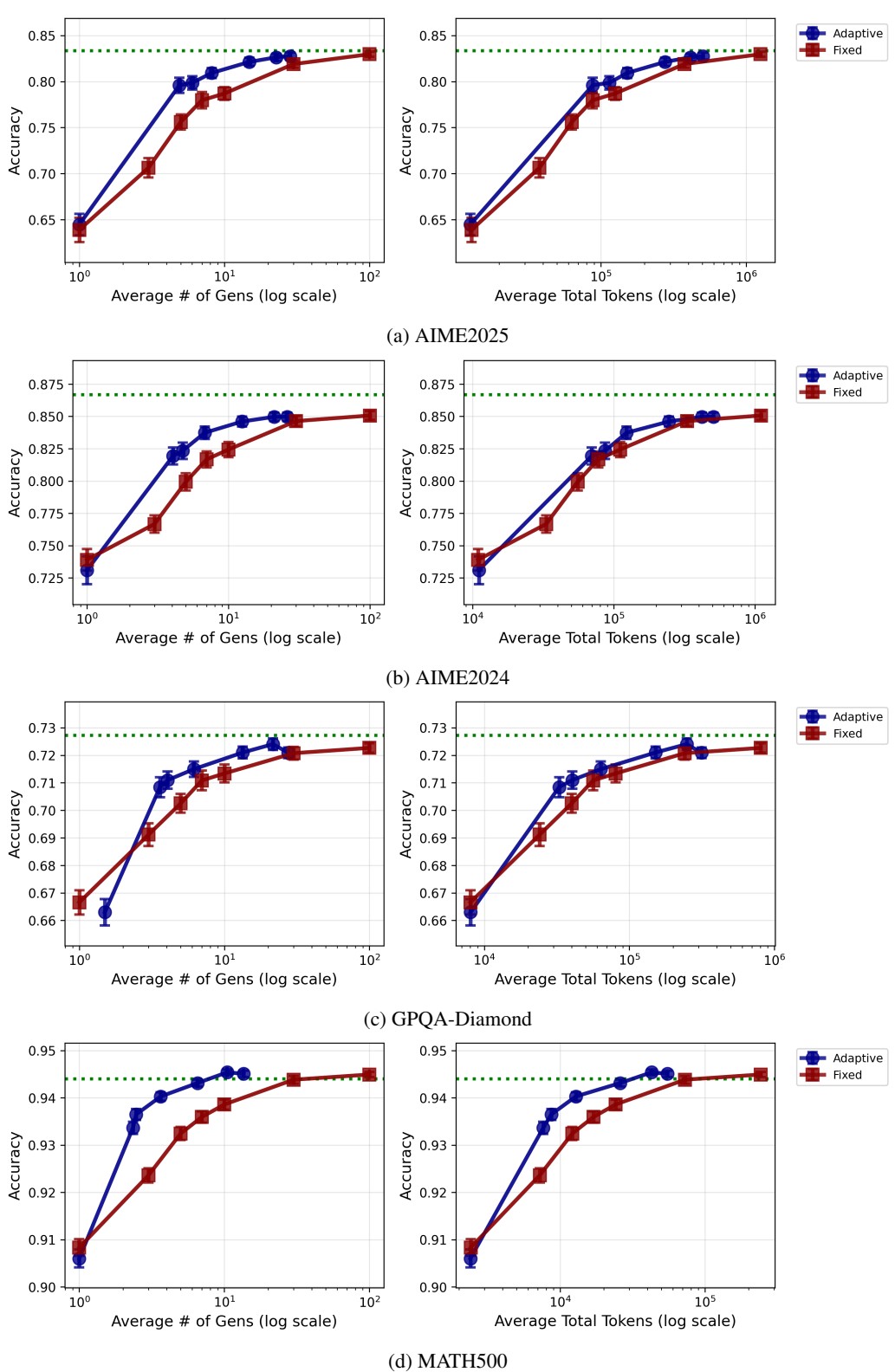

Figure 8: Cost-analysis of our proposed method and fixed BoN for Phi-4-reasoning. The error bars are standard two-sigma confidence intervals. Green dashed line indicates the best-of-$\infty$ performance.

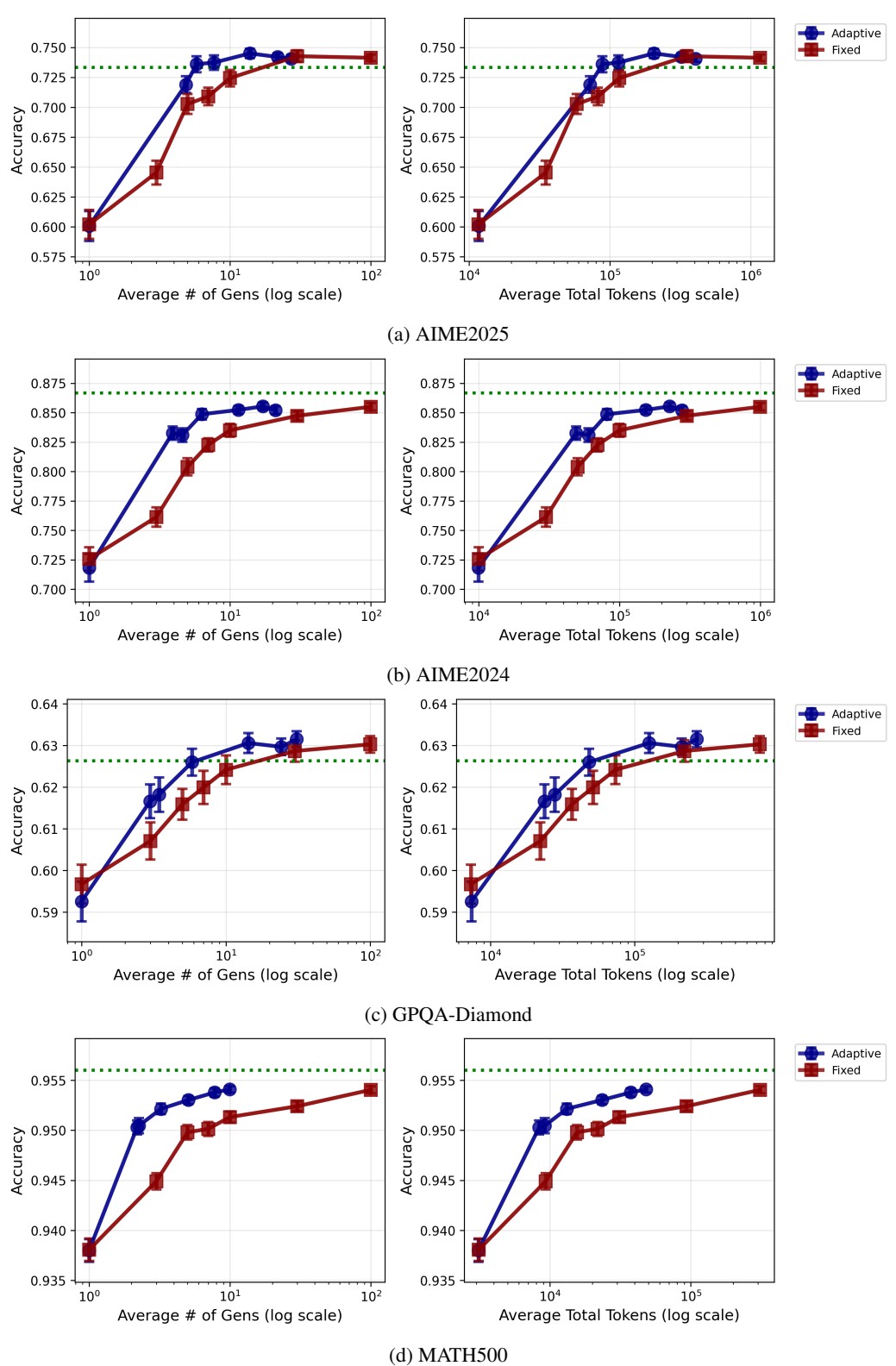

Figure 9: Cost-analysis of our proposed method and fixed BoN for NVIDIA-Nemotron-Nano-9B-v2. The error bars are standard two-sigma confidence intervals. Green dashed line indicates the best-of-$\infty$ performance.

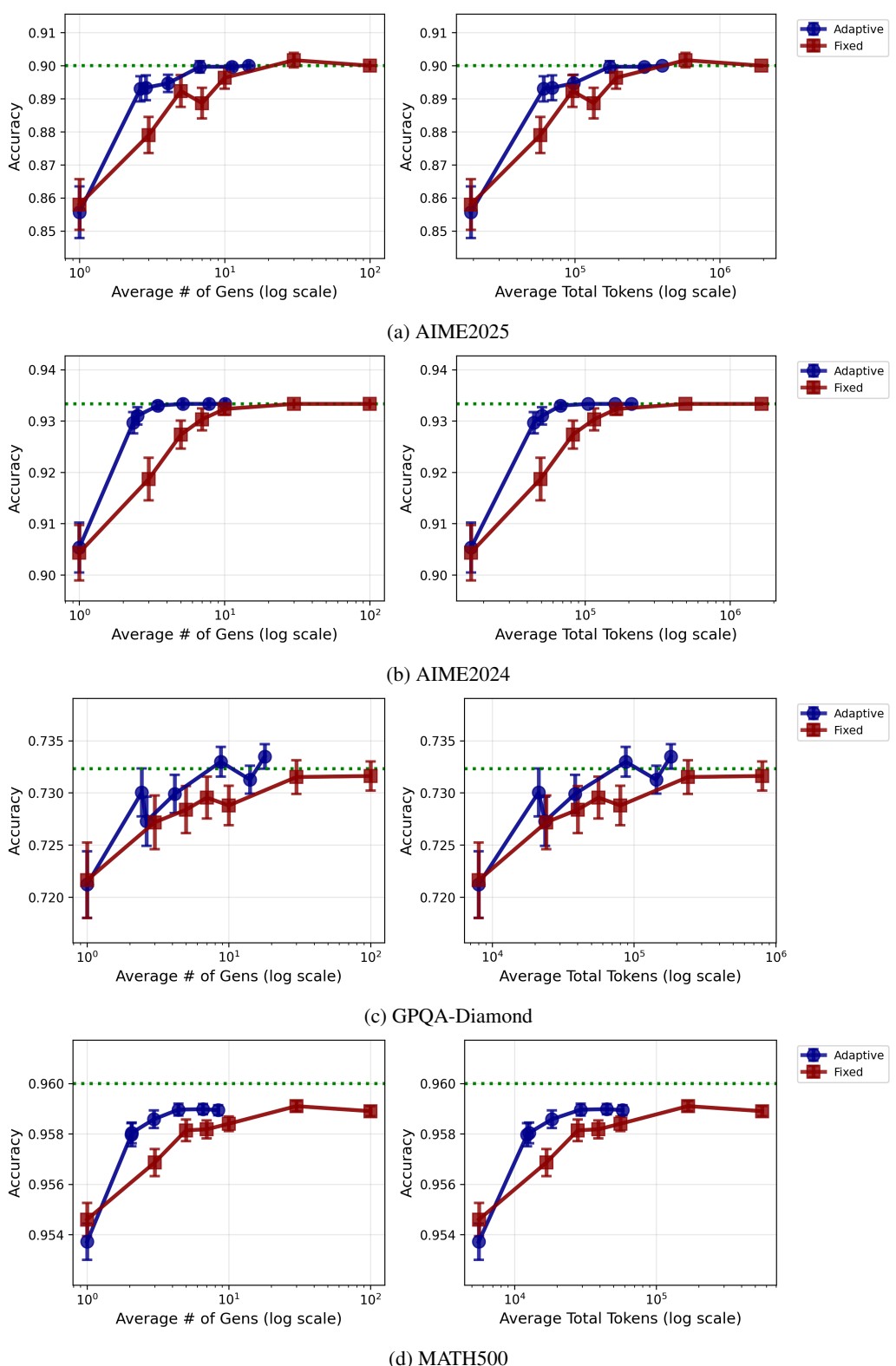

(a) AIME2025

(b) AIME2024

(c) GPQA-Diamond

(d) MATH500

Figure 10: Cost-analysis of our proposed method and fixed BoN for Qwen3-30B-A3B-Thinking-2507. The error bars are standard two-sigma confidence intervals. Green dashed line indicates the best-of-$\infty$ performance.

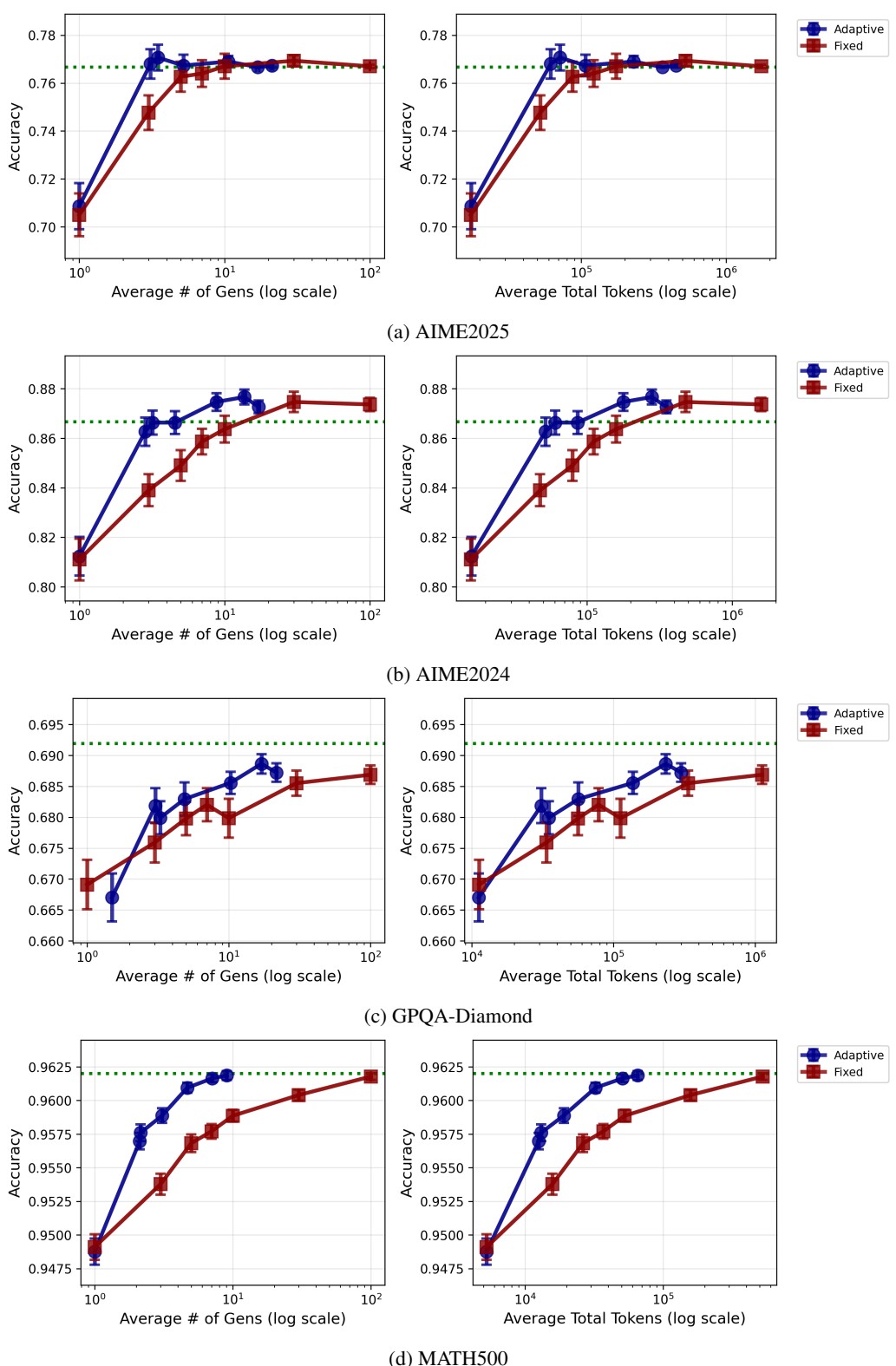

(a) AIME2025

(b) AIME2024

(c) GPQA-Diamond

(d) MATH500

Figure 11: Cost-analysis of our proposed method and fixed BoN for EXAONE-Deep-32B. The error bars are standard two-sigma confidence intervals. Green dashed line indicates the best-of-$\infty$ performance.

## G.2 EXPERIMENTAL SET 2: ADVANTAGE OF LLM ENSEMBLE OVER SINGLE LLM

Figure 12 demonstrates several more examples where the ensemble of LLMs outperforms the best single LLM. The weights are optimized by the MILP introduced in Section 3. We used Algorithm 1 to adaptively select and ask LLM for the answers.

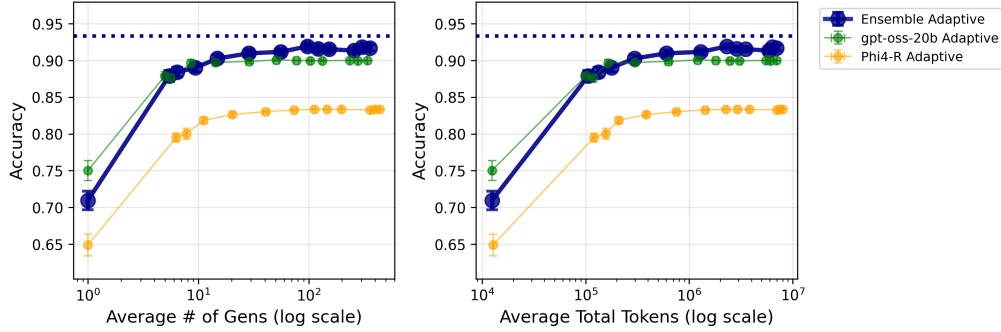

(a) Performance of a two-LLM ensemble. We used GPT-OSS-20B and Phi-4-reasoning on AIME2025. We tested with weight $w = (0.7, 0.3)$. The best-of-$\infty$ performance of GPT-OSS-20B is 0.900 (90.0%), whereas the ensemble's best-of-$\infty$ performance is 0.933 (93.3%).

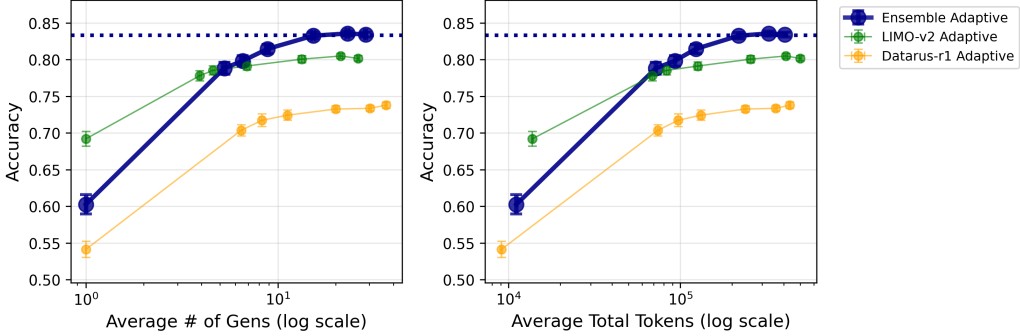

(b) Performance of two-LLM ensemble. We used LIMO-v2 and Datarus-R1-14B on AIME2024. The weight was optimized to $w = (0.4316, 0.5684)$.

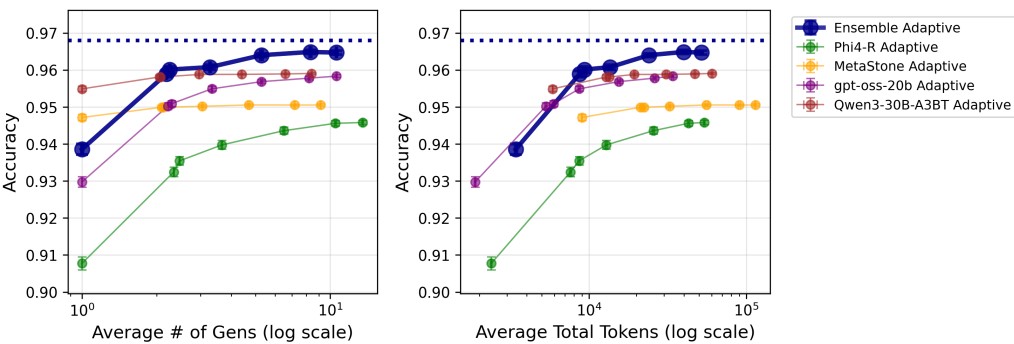

(c) Performance of four-LLM ensemble (MetaStone-S1-32B, Phi-4-reasoning, Qwen3-30B-A3B-Thinking-2507, and GPT-OSS-20B) on MATH500. The weight was optimized to $w = (0.0193, 0.0411, 0.3771, 0.5625)$.

Figure 12: Performance of LLM ensembles compared with single-LLM performance. We used Algorithm 1 choosing the LLM. Blue dashed line indicates the best-of-$\infty$ performance of the LLM ensemble.

## G.3 EXPERIMENTAL SET 3: LEARNING A GOOD WEIGHT

Figure 13 shows several additional examples of sample efficiency of learning the optimal weights in LLM ensembles. Dashed lines are the best-of-∞ performance of the individual LLMs. One can see that, with a small number of gold answers, the learned weights can outperform the best single LLM.

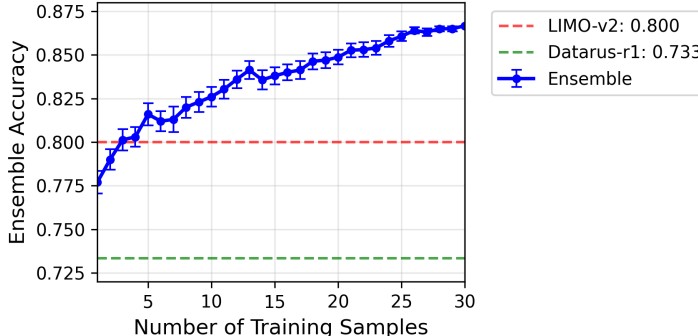

(a) The mixture of LIMO-v2 and Datarus-R1-14B on AIME2024. Note that the best-of-∞ performance of the two base LLMs is exactly the same and thus overlaps in the figure.

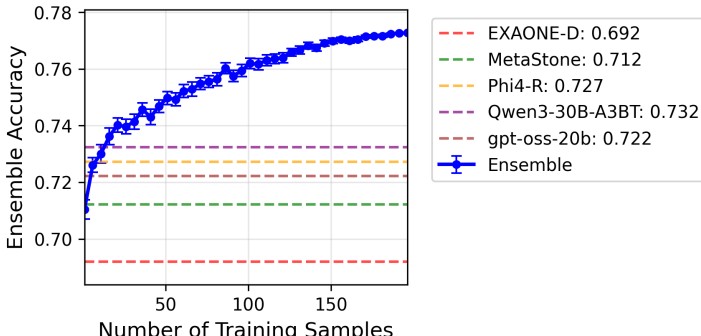

(b) The mixture of Phi-4-reasoning, Qwen3-30B-A3B-Thinking-2507, and GPT-OSS-20B on GPQA-Diamond.

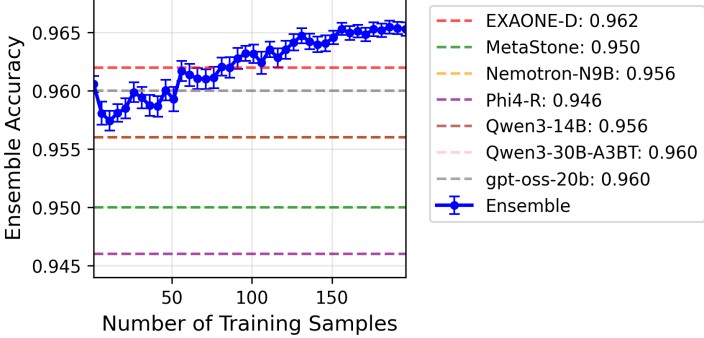

(c) The mixture of seven LLMS on MATH500.

Figure 13: The training of weights in an LLM ensemble. We show the number of samples to determine the weight (x-axis) versus the best-of-∞ performance (y-axis, i.e., performance of best-of-∞ with the weight). The $x$-axis indicates the number of problems used to learn the weight and the $y$-axis indicates the best-of-∞ performance. The score is averaged over 100 runs.

## G.4 EXPERIMENTAL SET 4: TRANSFER LEARNING OF THE OPTIMAL WEIGHT

We do not have additional experiments for this set of experiments.

## G.5 EXPERIMENTAL SET 5: COMPARISON WITH OTHER ANSWER-SELECTION METHODS

This section reports our comparison of majority voting with other aggregation methods. This appendix section complements Table 2 of main paper by providing additional experimental results on AIME2025 and other LLMs, as well as more details on the compared methods. The results are shown in Table 2. The compared methods are as follows:

- Omniscient is the hypothetical selection method that can always select the correct answer if it is included in the candidates, which is infeasible unless we know the gold answer. By definition, this is the best possible performance of any selection method.

- Majority voting is the method that selects the most frequent answer among the candidates. Ties are broken randomly.

- LLM-as-a-judge is the answer selection method that uses the target LLM itself to select the best answer among the candidates. Since the concatenation of the all answers can exceed the context length, we extracted the last 5,000 characters before the `</think>` tag of the answers for each answer.[11] To avoid uninterpretable answer, we ask the LLM twice, which slightly increased the accuracy. There are two variants: (tournament) compares the answers pairwise and selects the best one, and (set) compares all answers at once and selects the best one.

- INF-ORM-Llama3.1-70 is one of the state-of-the-art reward model (Minghao Yang, 2024), which marked the 9th in the RewardBench leaderboard as of September 8 2025.

- Skywork-Reward-V2-Llama-3.1-8B and Skywork-Reward-V2-Qwen3-8B are two of the state-of-the-art reward model (Liu et al., 2024), which marked the 1st and the 6th in the RewardBench leaderboard as of September 8 2025.

- Self-certainty is the method that selects the answer with the highest self-certainty score (Zhao et al., 2025b), which measures intrinsic confidence by how the likelihood differs from the uniform distribution per token. Note that we used the sequence average of self-certainty. Very recently, (Fu et al., 2025) introduced a version of self-certainty that weights more on the latter part of the sequence, which we have not tested and may improve the performance.

- Random is the model that randomly selects one of the candidates, whose performance should be close to the accuracy of a Bo1.

We use the same set of answers for comparing these selection methods, which reduces the variance due to the randomness in answer generation. Table 6, Table 7, and Table 8 show the comparison of these methods on GPT-OSS-20B, Phi-4-reasoning, and Qwen3-30B-A3B-Thinking-2507, respectively. The results are consistent with the main experiments in the paper. All results are Bo5 settings.

---

[11]For an answer without `</think>` tag, we used the final 5,000 characters. We also tested an alternative method that asks LLM to summarize its own answer before the comparison, which, in our preliminary analysis, did not outperform the proposed method.

Table 6: The accuracy of several selection methods on the best-of-five (Bo5) setting across three datasets (AIME2024, MATH500, GPQA-Diamond). Answers are generated by GPT-OSS-20B. The scores are averaged over 16 trials and we report the two-sigma confidence intervals.

| Method | AIME2024 | GPQA-Diamond | MATH500 |
|---|---|---|---|
| Omniscient | $91.25 \pm 1.03$ | $85.98 \pm 1.19$ | $95.56 \pm 0.23$ |
| Majority voting | $88.12 \pm 1.49$ | $70.07 \pm 2.02$ | $95.31 \pm 0.17$ |
| LLM-as-a-judge (set) | $85.42 \pm 1.48$ | $69.14 \pm 1.60$ | $94.31 \pm 0.28$ |
| LLM-as-a-judge (tournament) | – | $70.22 \pm 1.96$ | – |
| INF-ORM-Llama3.1-70B | $85.42 \pm 2.18$ | $68.38 \pm 1.84$ | $94.21 \pm 0.29$ |
| Skywork-Reward-V2-Llama-3.1-8B | $85.42 \pm 2.10$ | $68.13 \pm 1.95$ | – |
| Skywork-Reward-V2-Qwen3-8B | – | $68.42 \pm 1.93$ | – |
| Self-certainty | $81.67 \pm 2.98$ | $67.65 \pm 1.38$ | $93.50 \pm 0.47$ |
| Random ($\approx$ Bo1) | $79.17 \pm 2.89$ | $67.65 \pm 1.38$ | $93.91 \pm 0.40$ |

Table 7: The accuracy of several selection methods on the best-of-five (Bo5) setting on the AIME2025 and AIME2024 datasets. Answers are generated by Phi-4-reasoning. Scores are averaged over 16 trials and we report the two-sigma confidence intervals.

| Method | AIME2025 | AIME2024 |
|---|---|---|
| Omniscient | $85.00 \pm 1.72$ | $85.21 \pm 1.21$ |
| Majority voting | $76.67 \pm 2.58$ | $80.00 \pm 1.72$ |
| LLM-as-a-judge (set) | $72.92 \pm 3.10$ | $80.42 \pm 1.81$ |
| INF-ORM-Llama3.1-70B | $70.42 \pm 2.78$ | $78.54 \pm 2.51$ |
| Skywork-Reward-V2-Qwen3-8B | $70.62 \pm 2.87$ | $77.29 \pm 2.60$ |
| Self-certainty | $63.12 \pm 3.36$ | $73.54 \pm 2.31$ |
| Random ($\approx$ Bo1) | $63.96 \pm 2.45$ | $73.54 \pm 2.31$ |

Table 8: The accuracy of several selection methods on the best-of-five (Bo5) setting on the AIME2025 and AIME2024 datasets. Answers are generated by Qwen3-30B-A3B-Thinking-2507. Scores are averaged over 16 trials and we report the two-sigma confidence intervals.

| Method | AIME2025 | AIME2024 |
|---|---|---|
| Omniscient | $92.71 \pm 1.09$ | $93.54 \pm 0.74$ |
| Majority voting | $88.75 \pm 1.20$ | $92.92 \pm 0.57$ |
| LLM-as-a-judge (set) | $88.13 \pm 1.49$ | $92.29 \pm 0.80$ |
| LLM-as-a-judge (tournament) | $87.50 \pm 1.29$ | $91.25 \pm 1.48$ |
| INF-ORM-Llama3.1-70B | $89.38 \pm 1.09$ | $92.29 \pm 1.00$ |
| Skywork-Reward-V2-Qwen3-8B | $89.38 \pm 1.09$ | $92.71 \pm 0.67$ |
| Self-certainty | $87.50 \pm 2.06$ | $91.25 \pm 1.20$ |
| Random ($\approx$ Bo1) | $86.04 \pm 2.04$ | $90.00 \pm 1.36$ |

### G.6 COMPARISON WITH BETA STOPPING

In this section, we compare our proposed adaptive sampling algorithm (Algorithm 1) with the Beta stopping method introduced by Aggarwal et al. (2023), which is a state-of-the-art adaptive sampling method for best-of-$N$ (BoN) in LLMs. The Beta stopping method uses a Bayesian approach to determine when to stop sampling based on the posterior distribution of the majority and the second majority. Such a posterior projected to two-dimensional space follows a Beta distribution, and they uses the posterior probability such that the second majority exceeds the majority to decide whether to stop or not. Key findings are twofold. First, our proposed method dominates in the sense that our method is always as good as the Beta stopping. Second, there are several case where our method outperforms the Beta stopping. Results for GPT-OSS-20B, Phi-4-reasoning, EXAONE-Deep-32B are shown in Figure 14, Figure 15, and Figure 16, respectively.

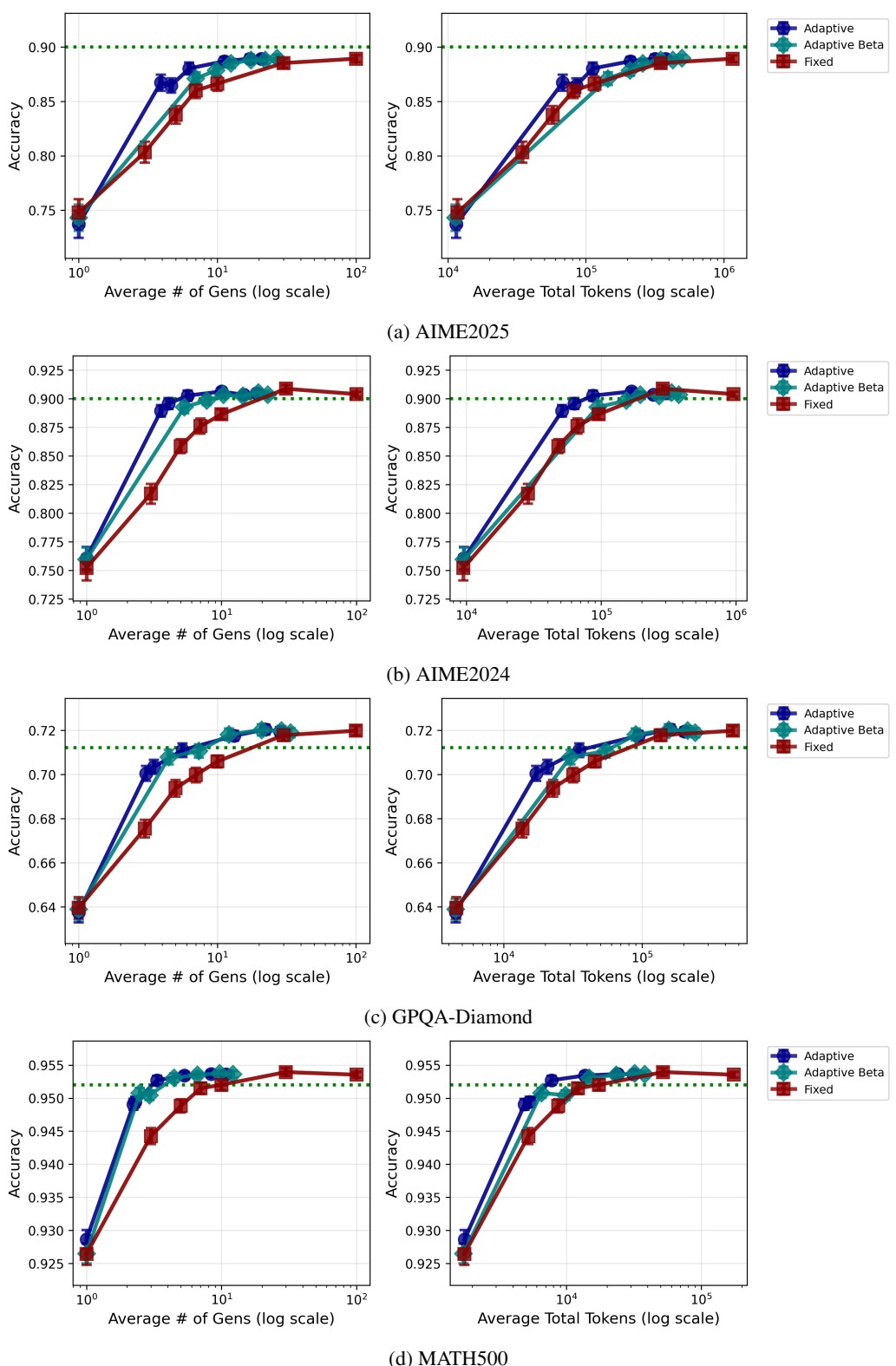

Figure 14: Cost-analysis of our proposed method ("adaptive"), Beta stopping (Aggarwal et al., 2023), and fixed BoN for GPT-OSS-20B. The error bars are standard two-sigma confidence intervals. Green dashed line indicates the best-of-∞ performance.

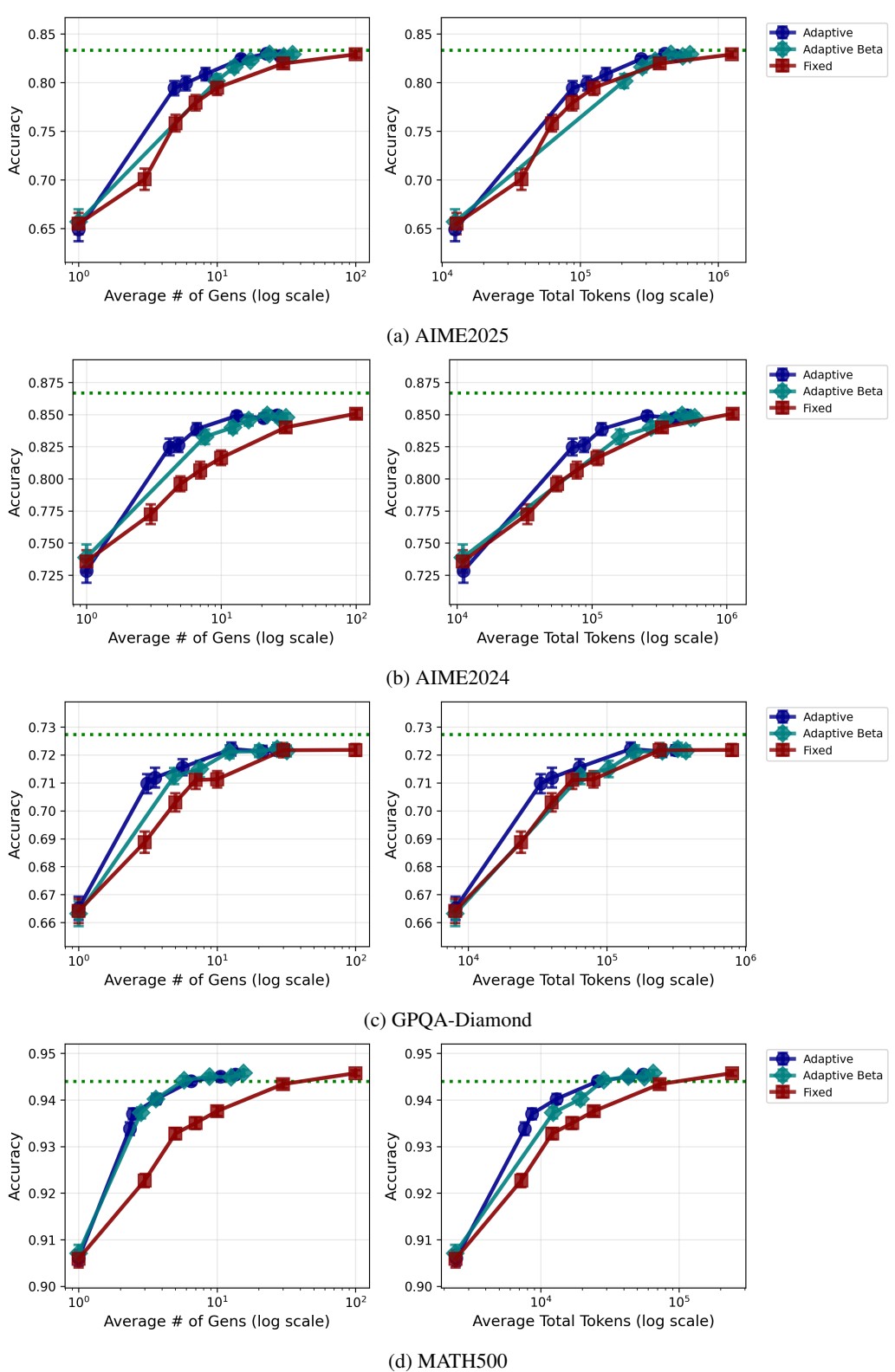

Figure 15: Cost-analysis of our proposed method ("adaptive"), Beta stopping (Aggarwal et al., 2023), and fixed BoN for Phi-4-reasoning. The error bars are standard two-sigma confidence intervals. Green dashed line indicates the best-of-$\infty$ performance.

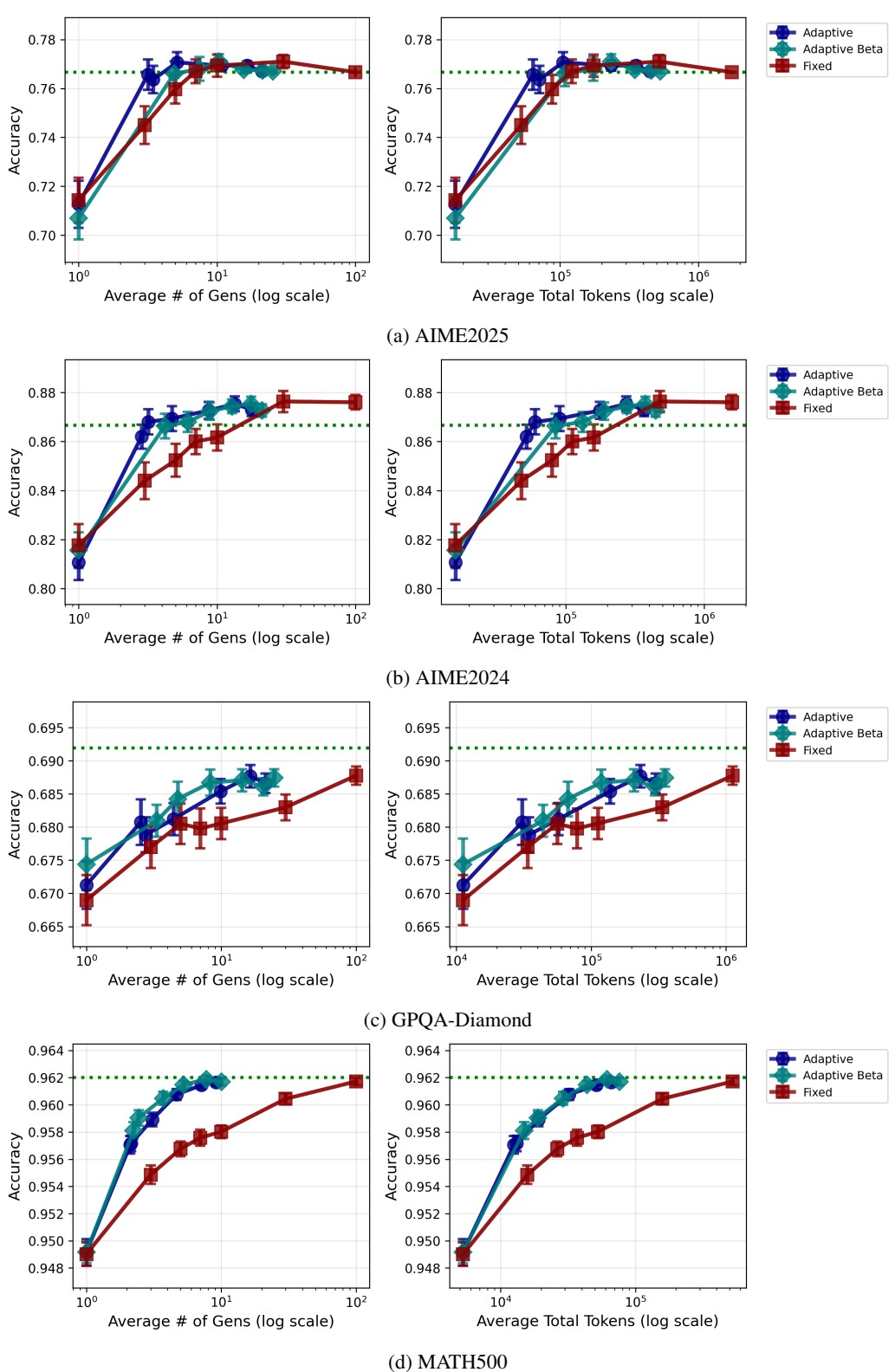

Figure 16: Cost-analysis of our proposed method ("adaptive"), Beta stopping (Aggarwal et al., 2023), and fixed BoN for EXAONE-Deep-32B. The error bars are standard two-sigma confidence intervals. Green dashed line indicates the best-of-∞ performance.

## H  ORACLE ADAPTIVE SAMPLER

Finally, we compare the performance of the proposed method with "oracle adaptive sampler" that stops when at least one of the samples are correct or reaches $N_{\max}$. Of course, such an adaptive algorithm requires an oracle (omniscient) verifier (Brown et al., 2024; Wu et al., 2024) that requires the knowledge of the gold answer and is unavailable in practice. Too see the performance of such oracle adaptive sampler, take an example of GPT-OSS-20B on the AIME2025 dataset (Figure 14 (a)). For example, an oracle adaptive sampler with $N_{\max} = 5$ has average $N \approx 1.6$ and accuracy $\geq 0.90$, which, by a large margin, exceeds the performance of the proposed algorithm. This is reasonable given that oracle does not need to depend on majority and one correct sample suffices to stop.

## I  ADDITIONAL DATASETS AND EXPERIMENTS

To further verify the robustness of the results, we tested two additional datasets.

The MedRECT dataset (Iwase et al., 2025) is a cross-lingual dataset that measures the ability of LLMs to detect and correct errors in clinical texts, where reasoning capability improves accuracy. In particular, we used the Japanese dataset[12] , which consists of 663 texts on medical diagnosis. Following Iwase et al. (2025), we extracted 367 texts that contain misdiagnoses and asked the LLM to detect the number of sentences that require correction.

The IMOBench dataset (Luong et al., 2025) contains 400 mathematical questions extracted from International Mathematical Olympiad (IMO) problems.[13] Unlike AIME, answers in this dataset may have multiple valid expressions (e.g., "1/2" is equivalent to "\frac12"), and our answer normalizer is imperfect. For evaluation, we used only the subset of questions for which at least one of the 100 generated answers matched the gold answer.

The results on these two datasets, which are consistent with our other experimental findings, are shown in Figure 17.

---

[12]https://github.com/pfnet-research/medrect/blob/main/data/medrect/medrect-ja-step4-accepted.json

[13]https://github.com/google-deepmind/superhuman/blob/main/imobench/answerbench_v2.csv

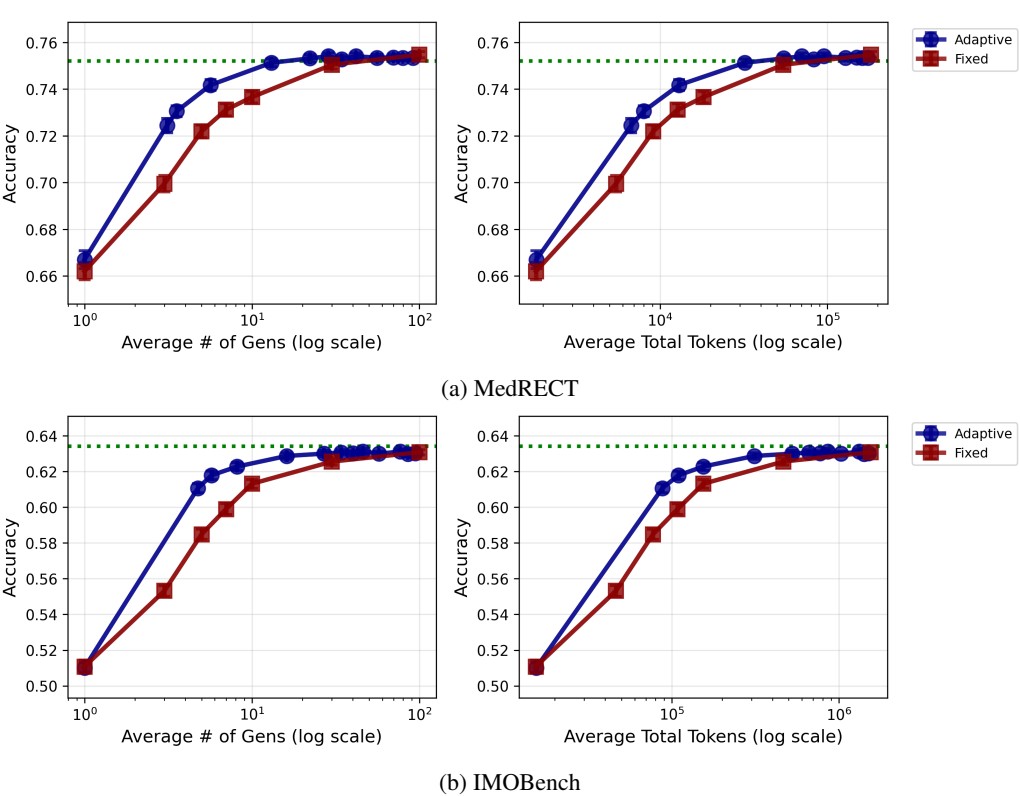

Figure 17: Cost-analysis of our proposed method and fixed BoN for GPT-OSS-20B. The error bars are standard two-sigma confidence intervals. Green dashed line indicates the best-of-∞ performance.

