# OpenReview forum: "Best-of-Infinity: Asymptotic Performance of Test-Time LLM Ensembling"
_ICLR.cc/2026/Conference — ICLR 2026 Poster_

### Official Review · Reviewer_jczD · 2025-10-20

**Soundness:** 2
**Presentation:** 3
**Contribution:** 2
**Rating:** 2
**Confidence:** 4

**Summary:**

This paper studies a test-time-scaling instance, where N answers are first generated by 1 LLM, and then one answer is selected by majority vote. The authors propose and analyze an algorithm that adaptively adjust number of answers to mitigate the overall LLM generation cost, which later is extended to the case where the N answers are generated by multiple LLMs (i.e., an LLM ensemble). Experiments using real-world LLMs are conducted to justify the results.

**Strengths:**

- Test-time-scaling is a timely topic, and its research impact may potentially last for a long time.


- The paper is well-written and easy to follow.


-The proposed majority vote approach is generic enough to be applied to a range of problems.

**Weaknesses:**

- The technical contributions are limited. The main contribution, Algorithm 1, simply views the LLM generated answers as samples from some unknown distributions and then estimates the uncertainty from the drawn samples to decide whether to go further or not. The theorems and lemmas all seem very straightforward too. See my questions for more details.

- Both majority-vote and LLM ensemble has been extensively studied in the literature. However, the paper ignores many (important) papers and fails to position itself appropriately. See my questions for more details.

- Motivations with open-source models: One main reason developers and researchers would use majority-vote is black-box access to the LLM models. Otherwise, instead of majority vote over multiple answers, one can simply fine-tune the model weights to optimize the performance. Hence, showing the performance with black-box LLM models (e.g., GPT-5 or at least GPT-5-mini) is essential to demonstrate the practical usefulness of Algorithm 1.

- The empirical results appear to be statistically insignificant and outdated due to dataset selections, and thus is hardly convincing that the proposed Algorithm 1 is efficient and effective.  See my questions below.

**Questions:**

- Where is the finite-sample analysis? Theorem 1 analyzes the asymptotic performance of Algorithm 1, i.e., showing it converges to best-of-infinity when the budget goes to infinity. This is not quite useful, as the standard majority-vote using N samples also converges to best-of-infinity as N goes to infinity. What is useful is the finite-sample analysis, i.e., the convergence rate of Algorithm 1 compared with majority-vote with the same number of samples.

- Why is the choice of Dirichlet process in Algorithm 1? As far as I understand, the purpose of Dirichlet process in Algorithm 1 is to offer a prior distribution for uncertainty estimation. There are several possible prior options, and I am not convinced that Dirichlet process is the simplest or optimal. At least one should provide some ablation study on the choices of the prior distribution.

- "The equation 8 is equivalent to the following MILP". How could an equation be the same as an optimization problem? Do the authors mean "maximizing equation 8"?

- Many closely related papers are completely ignored. For example, [J. Li et al 2025] and [L. Chen et al 2025] have already studies the case of majority vote with many generations and offered detailed analysis and empirical experiments. A. Huang studies how to optimize the best-of-n approach from a theoretical aspect. F. Wang studies almost the exactly same problem and proposes a similar algorithm to improve sample efficiency. The novelty and technical contributions cannot be easily accessed without clearly positioning the paper in the fruitful literature.

J. Li et al, More Agents Is All You Need, 2025.
L. Chen et al, Are More LLM Calls All You Need? Towards Scaling Laws of Compound Inference Systems, 2025.
A. Huang et al, Is Best-of-N the Best of Them? Coverage, Scaling, and Optimality in Inference-Time Alignment, 2025.
F. Wang et al, DynScaling: Efficient Verifier-free Inference Scaling via Dynamic and Integrated Sampling, 2025.

- The empirical results are based on only 3 datasets, namely, AIME(2024+2025), GPQA, and MATH500. While relevant, GPQA and MATH500 have been released for a while and often considered contaminated. AIME (2024 or 2025), on the other hand, only contains 30 questions, making it extremely hard to draw any significant conclusions. Large-scale experiments are essential for statistically significant conclusions.

- What is the training runtime cost? Obtaining the optimal weights for LLM ensemble is via solving an MILP, which can be extremely expensive. What solvers are used in the experiments? How fast/slow are them?

---

> ### Author Response · Authors · 2025-11-19
>
> We thank the reviewer for the insightful comments. It would be great if you also see "Overall Response" above.
>
> > Motivations with open-source models: ... , showing the performance with black-box LLM models (e.g., GPT-5 or at least GPT-5-mini) is essential to demonstrate the practical usefulness of Algorithm 1.
>
> We gracefully disagree on this point. Many proprietary companies use open-weight models because it is more affordable and have reasonable performance than frontier models like GPT-5, and thus, empirical demonstration with open-weight models is very important. We did not choose frontier models for the following reasons. First, their reproducibility is not guaranteed, as companies can discontinue with them or update with new ones at any time. Second, we are not willing to depend on a specific proprietary product. Finally, many works such as [1,2] exclusively work on open-weight LLMs.
>
> > Where is the finite-sample analysis? Theorem 1 analyzes the asymptotic performance of Algorithm 1, i.e., showing it converges to best-of-infinity when the budget goes to infinity. This is not quite useful, as the standard majority-vote using N samples also converges to best-of-infinity as N goes to infinity. What is useful is the finite-sample analysis, i.e., the convergence rate of Algorithm 1 compared with majority-vote with the same number of samples.
>
> We agree that Theorem 1 is weak. Happily, we are expert on frequentist finite-time analysis of Bayesian models, and we have updated the paper to add the stopping time that is $O(\log(1/B)/\Delta^2)$ to the Bayes factor $B$ and $\Delta$ (gap between majority and second majority) at the new section "C.1 FINITE-TIME ANALYSIS OF STOPPING TIME". The implication of this is that $1/B$ can be interpreted as the frequentist confidence level. The results use the fact that the Dirichlet distribution projected into two dimensions is the Beta distribution and include somewhat unknown results on Beta concentration (Lemma 5 therein), which is novel to the community and makes our paper theoretically interesting. That said, the Bayesian method is harder than confidence-bound methods such as UCB, and our characterization is only on the stopping time. If you consider this a critical point, we can even attempt to add further characterization of the method's accuracy.
>
> > Why is the choice of Dirichlet process in Algorithm 1? As far as I understand, the purpose of Dirichlet process in Algorithm 1 is to offer a prior distribution for uncertainty estimation. There are several possible prior options, and I am not convinced that Dirichlet process is the simplest or optimal. At least one should provide some ablation study on the choices of the prior distribution.
>
> We believe the Dirichlet process is the primary choice when we consider distributions for which the candidate set is unknown. Regarding the choice of prior, the Dirichlet process is a "flat" prior. In a slightly different context, a Beta distribution with a flat prior works in sequential decision-making [3]. If we jointly estimate the mean and variance, there may be some sensitivity to the prior.
>
> > Many closely related papers are completely ignored. For example, [J. Li et al 2025] and [L. Chen et al 2025] have already studies the case of majority vote with many generations and offered detailed analysis and empirical experiments. A. Huang studies how to optimize the best-of-n approach from a theoretical aspect. [F. Wang] studies almost the exactly same problem and proposes a similar algorithm to improve sample efficiency. The novelty and technical contributions cannot be easily accessed without clearly positioning the paper in the fruitful literature.
>
> Thank you for your suggestions. Our main contribution concerns the ensemble LLM and its performance limits that is not limited to single-LLM. We have revised the paper to cite these works.
>
> > The empirical results are based on only 3 datasets, namely, AIME(2024+2025), GPQA, and MATH500. While relevant, GPQA and MATH500 have been released for a while and often considered contaminated. AIME (2024 or 2025), on the other hand, only contains 30 questions, making it extremely hard to draw any significant conclusions.
>
> Thank you for your suggestions. We will add several datasets that were generated in 2025 to the camera-ready.
>
> > What is the training runtime cost? Obtaining the optimal weights for LLM ensemble is via solving an MILP, which can be extremely expensive. What solvers are used in the experiments? How fast/slow are them?
>
> Each MILP finished in at most a few seconds. Compared with LLM's generation, the cost of MILP is insignificant.
>
> - [1] A. Huang et al, Is Best-of-N the Best of Them? Coverage, Scaling, and Optimality in Inference-Time Alignment, 2025.
> - [2] Deep Think with Confidence. Yichao Fu, Xuewei Wang, Yuandong Tian, Jiawei Zhao.
> - [3] Optimality of Thompson Sampling for Gaussian Bandits Depends on Priors. Junya Honda, Akimichi Takemura.

---

### Official Review · Reviewer_XSCw · 2025-10-25

**Soundness:** 3
**Presentation:** 2
**Contribution:** 2
**Rating:** 6
**Confidence:** 4

**Summary:**

This paper studies adaptive repeated sampling methods for more compute-efficient test-time scaling. This paper also proposes a new approach to learning optimal weighting for multi-LLM ensemble based on mixed-integer linear programing.

**Strengths:**

- The research topic is important and well motivated.

- The proposed methods are grounded with principled analysis.

- The experiments show that the proposed methods lead to performance improvement.

**Weaknesses:**

## Main weaknesses
- I find some terminologies in the paper confusing or inappropriate:
  - Test-time compute: Whenever an LLM is used in the test-time, the compute should be categorized as test-time compute. Based on my understanding, this submission indeed focuses on test-time scaling (TTS), or different methods of scaling up test-time compute. Therefore, test-time scaling should be the right term to use (e.g., in the title), and this terminology has been widely adopted in existing papers [1].
  - Best-of-$N$: In early works on test-time scaling, best-of-$N$ specifically refers to selecting the best generation trajectory with an external verifier -- either an oracle or a learned verifier [2,3]. "Majority voting", to the best of my knowledge, is a different method from best-of-$N$ in existing studies [3,4]. After all, majority voting cannot select the single "best" response from $N$ candidates -- it just returns an output answer based on consensus. The focus of this paper is purely on majority voting and I cannot agree that this is a "variant" of "best-of-$N$". Besides, I think [2,3,4] -- [4] in particular -- are all very relevant to this work and should be discussed.

[1] s1: Simple test-time scaling

[2] Large Language Monkeys: Scaling Inference Compute with Repeated Sampling

[3] Inference Scaling Laws: An Empirical Analysis of Compute-Optimal Inference for Problem-Solving with Language Models

[4] Are More LLM Calls All You Need? Towards Scaling Laws of Compound Inference Systems

- The exposition of the method part lacks clarity:
  - Algorithm 1 is not even self-contained. The input $\alpha$ is not explicitly used. Moreover, the approximation of $\mathrm{BF}(n)$ lies in the center of the analysis in Section 2. The algorithm needs to elaborate on it rather than calling the function without any explaination.
  - Line 182 mentions that $\mathbb{P}(H_1\mid \mathcal D (n))$ can be estimated using Monte Carlo methods. More details on the method are anticipated, especially when the answer space is large or even infinite.
  - The "algorithm" in Theorem 1 is very vague. Is it the presented Algorithm 1? Is $\mathrm{BF}(n)$ approximated? Should the result also rely on a large number of Monte Carlo samples?
  - In Equation (8), the notation is wired. People naturally expect $f(\\{a_q\\})$ is defined for a single problem $q$ but it is indeed for the entire problem set. Why not denote it by $f{\mathcal Q)$?


## Nits
- Lines 28-30: `\citet` $\to$ `\citep`.
- Line 175: When $n$ is sufficiently large compared with $\alpha$ $\to$ When $n\gg \alpha$.
- Line 201: AIME problem solutions can be 0.
- Line 222: The "=" should be $\in$ since the maximizer is not necessarily unique.

**Questions:**

- Can you give some examples of the base distributions $H$ when the answer space is infinite?

- The MILP in Lemma 4 assumes access to $p_{i,j}^q$. How is that implemented in practice?

---

> ### Author Response · Authors · 2025-11-19
>
> We thank the reviewer for the insightful comments. Below, we answer each of your questions.
>
> > Test-time compute (terminology)
>
> Thank you for your suggestion. We consider the test-time compute is appropriate because our focus on the limit performance of test-time compute rather than scaling (it would be great if you also see "Overall Response" above).
>
> > Best-of-$N$ (terminology)
>
> We consider majority voting as a special case of BoN, as you and Reviewer 6Yj4 correctly noted. We can change the title and discussion if you think it would be more appropriate to separate them. In that case, our title would be like "Towards the Infinite Limit of Majority Voting for LLM Ensembles".
>
> Thank you for suggestion of related work. We will add the works [2,3,4] to the paper ([1] is already there).
>
> > The exposition of the method
>
> Thank you for your suggestion. We will revise the notation to improve the paper. Monte Carlo method samples from the Dirichlet distribution, where the $\alpha$ fraction is just discarded. BF$(n)$ is approximated. We will use ${a_q}$-underscore-${q \in \mathcal{Q}}$ (sorry, underscore does not work perfectly in OpenReview markdown) to represent answers to a problem set $\mathcal{Q}$, which is more standard.
>
> > Can you give some examples of the base distributions $H$ when the answer space is infinite?
>
> For example, $H$ can be probability distribution over natural numbers $1,2,3,\dots$ where the probability of $n$ is $(1/n^2) / Z$, where $Z = \sum_n 1/n^2 = \pi^2/6$ is the normalizer.
>
> > The MILP in Lemma 4 assumes access to $p_{i,j}^q$. How is that implemented in practice?
>
> We use a plug-in estimate based on all our generations.
>
> Thank you for your other suggestions. We will improve the paper accordingly.

---

### Official Review · Reviewer_6Yj4 · 2025-11-01

**Soundness:** 2
**Presentation:** 2
**Contribution:** 2
**Rating:** 4
**Confidence:** 3

**Summary:**

This work investigates test-time scaling of large language models (LLMs) via majority voting. The authors analyze the infinite-sample limit of accuracy and propose an instance-level dynamic compute allocation method based on answer agreement. They also consider weighted ensembles of multiple LLMs and formulate a mixed-integer linear program to determine optimal weights. Experiments are conducted to validate the proposed analysis and methods.

**Strengths:**

- The proposed dynamic compute allocation method based on answer agreement is sensible, intuitive, and straightforward to implement.
- The release of a dataset containing LLM-generated responses may be useful to researchers studying LLM test-time scaling.

**Weaknesses:**

- The procedure for optimizing the weights in the LLM ensemble is unclear. In particular, it appears that the same data is used for both optimization and evaluation, which would amount to "training on test data" and thus compromise the validity of the results.
- Some theoretical results (e.g., Theorem 1 and Lemma 1) are relatively obvious and could be better presented by briefly summarizing the key insights in the main text, with full details and formal notation moved to the appendix.
- "Experiment Set 5" on page 9 seems out of place, as it compares standard majority voting with other test-time scaling methods, none of which is directly related to the proposed methodology or analysis.
- The discussion on asymptotic performance may give the impression that majority voting's accuracy increases monotonically with the number of samples, which is not always correct. This paper misses an important related work [1] demonstrating that the accuracy curve can exhibit diverse behaviors.

  [1] Are More LLM Calls All You Need? Towards the Scaling Properties of Compound AI Systems (NeurIPS 2024)

- Minor issues:
  - Line 184: replace “the algorithm” with “Algorithm 1.”
  - Footnote 4 on page 5 seems redundant and could be removed.
  - Table captions should be placed above the tables, per ICLR format guidelines.
  - Line 482 contains a grammatical error: “we have verified robust across LLMs and problem sets”.
  - Table 6 has missing entries without explanation.

**Questions:**

- How is asymptotic accuracy computed in the experiments? In the first plot of Figure 1, there is a noticeable gap between asymptotic accuracy and empirical accuracy at $N$ as large as 1000; how should this be interpreted?
- Regarding terminology: literature convention generally considers majority voting and best-of-N as separate methods (as far as I can tell), whereas this paper treats majority voting as a special case of best-of-N. Nonetheless, I do not object strongly to this choice if other reviewers think this is fine.

---

> ### Author Response · Authors · 2025-11-19
>
> We thank the reviewer for the insightful comments. Below, we answer each of your questions. It would be great if you also see "Overall Response" above.
>
> > The procedure for optimizing the weights in the LLM ensemble is unclear. In particular, it appears that the same data is used for both optimization and evaluation, which would amount to "training on test data" and thus compromise the validity of the results.
>
> Determination of the optimal weight in the case of LLM ensemble requires the gold answer as you correctly suggested. In practice, knowing a small fraction of gold answers suffices. In particular, Figure 6 shows that in AIME2025, only 5 gold answers achieve the weight as good as the best LLM among eleven LLMs. Additional results with other LLMs and other datasets are provided in the supplementary material "Experimental Set 3."
>
> > The discussion on asymptotic performance may give the impression that the majority voting's accuracy increases monotonically with the number of samples, which is not always correct. This paper misses an important related work [1] demonstrating that the accuracy curve can exhibit diverse behaviors.
>
> We appreciate your suggestion. We agree that the Best-of-$\infty$ performance can be smaller than finite-sample performance in some cases. However, paper [1] as well as our paper show that the majority vote overall improves on the accuracy, but we will clarify that the accuracy can decrease with a larger n. [1] also consider reasoning favors majority voting, which is consistent with our view. We revise the paper to add a citation to [1].
>
> > Minor issues
>
> Thank you for your suggestions. We will revise the paper accordingly.
>
> >  How is asymptotic accuracy computed in the experiments? In the first plot of Figure 1, there is a noticeable gap between asymptotic accuracy and empirical accuracy at $N$ as large as 1000; how should this be interpreted?
>
> Asymptotic accuracy is based on majority voting on all of our empirical data. The gap in Fig 1 AIME2024 is based on the following reason. Namely, AIME2024 has 30 questions (one question is 3.3% of the accuracy) and the majority is incorrect, and the second majority, which is very close to the majority, is correct. In this case, Best-of-$\infty$ on this fraction slowly goes to 0, whereas finite samples obtain some fraction of 3.3%.
>
> > Regarding terminology: literature convention generally considers majority voting and best-of-N as separate methods (as far as I can tell), whereas this paper treats majority voting as a special case of best-of-N. Nonetheless, I do not object strongly to this choice if other reviewers think this is fine.
>
> We consider majority voting as a special case of BoN, as you correctly noted. But we can change the title and discussion if you and Reviewer XSCw consider it more appropriate to separate.
>
> [1] Are More LLM Calls All You Need? Towards the Scaling Properties of Compound AI Systems (NeurIPS 2024)

---

### Official Review · Reviewer_CXnE · 2025-11-04

**Soundness:** 3
**Presentation:** 3
**Contribution:** 2
**Rating:** 2
**Confidence:** 3

**Summary:**

The apparel provides an early stopping mechanism for sampling in the majority-voting algorithm in the LLMs to achieve the same performance as the infinite-budget case with finite generations. The mechanism maintains a Drichlet Process posterior over output distribution and stops according to the Bayes factor. Also, the paper studies the optimal weighting for LLM ensembles in the infinite budget case.

**Strengths:**

The paper is written rigorously with an accurate language. The paper is mostly easy to follow.

The introduced early stopping mechanism can be very impactful in practice

I find the ensemble problem very interesting and novel.

The experiments are extensive and cover many models and benchmarks. The algorithms work well across all settings.

**Weaknesses:**

I mostly have two major concerns about the paper;

1. It appears to me that the Drichlet stopping criteria highly resembles the work of [1]. I am not aware of the full details of that paper and would appreciate a comparison by the authors. However, it looks like both model the problem as a dirichlet process and stop using a threshold. This reduces the novelty of this part of the paper.

2. The ensemble method seems to depend on the knowledge of the ground truth answers. The matrix in the constraint in equation 13 depends on the ground truth answer. If this is true, it severely reduces the practicality of this part of the paper.


More minor issues:

3. In a footnote, It is mentioned that the method can be applied to infinite possible final answers (which is the case in MATH500) problems. But it breaks the approximation in equation 6. More clarification about the applicability to this setting strengthens the paper.

4. As a reference point, it would be informative to compare the stopping criteria with an oracle one. The oracle could be stopping at the moment the output becomes and remains correct.



[1] [Let’s Sample Step by Step: Adaptive-Consistency for Efficient Reasoning and Coding with LLMs](https://aclanthology.org/2023.emnlp-main.761/) (Aggarwal et al., EMNLP 2023)

**Questions:**

1. I could not follow the line 117 derivation and would appreciate more explanations. There might be typo there. To me, it should be

$$
\mathbb{P}(H_1 \mid \mathcal{D}(n)) \approx \Pr[X_1 \ge \max_{i \ne 1} X_i,\, X \sim \mathrm{Dirichlet}(N_1, N_2, \ldots, N_{s(n)}, \alpha)].
$$

Also, the assumption is that the probability of all $A_i$ together (instead of just A1) is zero under H.

---

> ### Author Response · Authors · 2025-11-19
>
> We thank the reviewer for the insightful comments. Below, we answer each of your questions. It would be great if you also see "Overall Response" above.
>
> Major issues
>
> > It appears to me that the Drichlet stopping criteria highly resembles the work of [1]. I am not aware of the full details of that paper and would appreciate a comparison by the authors. However, it looks like both model the problem as a dirichlet process and stop using a threshold. This reduces the novelty of this part of the paper.
>
> Thank you for your suggestion. There are two differences between our work and theirs. First, their work approximates the Dirichlet posterior with the Beta posterior of the top-2 majority. Second, our work is based on the Bayes factor that bonuses the number of appeared answers. We revise the paper to add the suggested work. Our new Section G.6 "Comparison with Beta stopping" shows the performance of our stopping almost dominates the stopping of [1] and significantly outperforms them in many cases.
>
> > The ensemble method seems to depend on the knowledge of the ground truth answers. The matrix in the constraint in equation 13 depends on the ground truth answer. If this is true, it severely reduces the practicality of this part of the paper.
>
> You are right that determination of the optimal weight requires the gold answer. In practice, knowing a small fraction of gold answers suffice. In particular, Figure 6 shows that in AIME2025, only 5 gold answers achieve the weight as good as the best LLM among eleven LLMs. Additional results with other LLMs and other datasets are provided in the supplementary material "Experimental Set 3."
>
> Minor issues
>
> > In a footnote, It is mentioned that the method can be applied to infinite possible final answers (which is the case in MATH500) problems. But it breaks the approximation in equation 6. More clarification about the applicability to this setting strengthens the paper.
>
> Thank you for the suggestion. Even though the MATH500 allows possibly infinite variety of answers, what matters here is the variety of possible answers by LLMs. If LLMs variation of answers is (effectively) finite, we consider it valid.
>
> > As a reference point, it would be informative to compare the stopping criteria with an oracle one. The oracle could be stopping at the moment the output becomes and remains correct.
>
> Thank you for your suggestion. We will add the suggested oracle in the camera-ready because this is marked as a minor point and we focus on conducting additional experiments on the major point (i.e., comparison with [1]). However, if you consider this a critical, we are happy to add one.
>
> Question
>
> > I could not follow the line 117 derivation and would appreciate more explanations.
>
> Thank you for your question. We appreciate your correction and will revise the paper as well as the experiment in the camera-ready.

---

### Author Response · Authors · 2025-11-19
**Overall Response**

Dear Reviewers and AC,

We appreciate the reviewer’s comments. We would like to respectfully point out that the review focuses primarily on the adaptive stopping aspect, which is only a part of our contributions. Our main contribution lies in characterizing the best-of-$\infty$ (infinite-sample) performance of LLM ensembles and formulating it via MILP optimization. Adaptive sampling is introduced only as a practical method to approximate this infinite-sample performance under finite computation.

Importantly, most of the suggested citations are about a single LLM and none of them provides such a characterization of the infinite-sample limit on the LLM ensemble, which implies the novelty of our work.

We answer each of the reviewers' questions in the corresponding replies.

---

> ### Author Response · Authors · 2025-11-19
> **Revision**
>
> Dear reviewers and AC,
>
> We have revised the paper. The major changes are as follows.
>
> * Cited the papers suggested by the reviewers.
> * As per the request by Reviewer CXnE, we have added additional comparison with the adaptive stopping method by [1].
> * As per the request by Reviewer jczD, we have added a finite-time analysis.
> * To emphasize our contribution to LLM ensembles, we have rewritten the introduction. We can further update it and the title if reviewers prefer.
>
> We hope this revision clarify our contributions.
>
> [1] Let’s Sample Step by Step: Adaptive-Consistency for Efficient Reasoning and Coding with LLMs (Aggarwal et al., EMNLP 2023)

---

### Author Response · Authors · 2025-12-04
**Summary of our results**

Dear AC,

We thank the reviewers for their constructive and thoughtful comments. We summarize our response:

* **Clarified our main contributions include LLM ensemble results:**
    * While several reviews focused primarily on the adaptive stopping component, we would like to kindly emphasize that this is only one part of our work. Our main contribution is the characterization of the best-of-$\infty$ (infinite-sample) performance of LLM ensembles and its formulation via a MILP optimization. The adaptive sampling method is introduced solely as a practical approach to approximate this infinite-sample performance under finite computational budgets.
* **Addressed the key concerns raised by reviewers by revision:**
   * In response to Reviewer CXnE’s helpful suggestion, we added a direct comparison to the adaptive sampling method of Aggarwal et al. (Section G.6, “Comparison with Beta stopping”). Our experiments show that our stopping rule almost always dominates theirs and often provides substantially better performance.
   * Following Reviewer jczD’s request for a finite-time analysis, we added Section C.1 (“Finite-Time Analysis of Stopping Time”), which clarifies the connection between our Bayes-factor stopping rule and frequentist confidence.
* **Release of a large CoT-generation dataset:**
   * Finally, we want to point out that we have released the dataset used in our experiments. It is larger than those in most existing works and we hope it will be a valuable resource for the community.

[1] Let’s Sample Step by Step: Adaptive-Consistency for Efficient Reasoning and Coding with LLMs (Aggarwal et al., EMNLP 2023)

We hope this clarifies the contributions. We appreciate your time and effort.

Sincerely,

Authors

---

### Meta-Review · Area_Chair_pKk3 · 2026-01-07

**Summary:**

Overall, the initial reviews seemed divided, with a strong reject, a weak reject and a weak accept. The main concerns focused on presentation and the relationship with the literature. While I am not an expert on the topic, I think the authors did a good job at rebutting the criticisms. It is a pity that no author-reviewer discussion was had: it would have been helpful to know whether the reviewers found the responses satisfying, especially regarding the relationship with the literature.

Given the circumstances, I'm marking this as a borderline paper with a weak accept recommendation.

**Reviewer Concerns:**

Most of the concerns focused on the presentation, relationship with the literature, and the model.

The presentation concerns, including on terminology, seem to have been mostly resolved.

The authors responded to the other concerns in their response, but it is unclear whether the reviewers would have found the answers satisfying. I personally found the responses reasonable, but I am not an expert on the topic.

**Reviewer Scores:**

It is hard to predict how the reviewers would have changed their score, given that only initial reviews and rebuttals were posted.

---

### Decision · Program_Chairs · 2026-01-26

Accept (Poster)